# ParaSet: Retrieving Set of Passages in an Efficient Manner

## Abstract

Multi-hop question answering and retrieval-augmented reasoning require selecting a query-dependent set of passages whose usefulness is determined by their joint compatibility, rather than by independently scoring passages. Most existing approaches, however, do not explicitly model query–set compatibility, relying instead on independent passage relevance or step-wise retrieval, which often results in locally optimal and brittle retrieval chains. We argue that holistic set-level compatibility learning is essential, yet directly enumerating and scoring passage sets lacks computational scalability. To bridge this gap, we propose `ParaSet` (Parallel-Set scorer) with two components. First, we introduce a set-level compatibility learning objective that enables retrievers to distinguish compatible and incompatible passage sets, yielding robust reasoning over variable-length and partially corrupted contexts. Second, we design a lightweight single-layer self-attention scorer, ParaSet, trained with the same objective, which enables efficient selection of promising passage sets. Together, our results demonstrate that set-level compatibility learning is effective for multi-hop question answering, and that ParaSet enables efficient exploration of the combinatorial passage-set space, leading to stable retrieval behavior and strong end-task performance on higher-hop questions.

## 1. Introduction

Document retrieval has traditionally focused on selecting the single most relevant passage for a given query. However, many reasoning-intensive tasks such as multi-hop QA require retrieving a *set of evidence passages*, where the relevance of each passage is inherently dependent on the others. In such settings, retrieving individually relevant passages is insufficient: a passage that is useful in isolation may become uninformative or even misleading when combined with an incompatible context. This interdependence makes multi-hop retrieval fundamentally a *set-level* problem, rendering standard single-passage retrieval approaches inadequate.

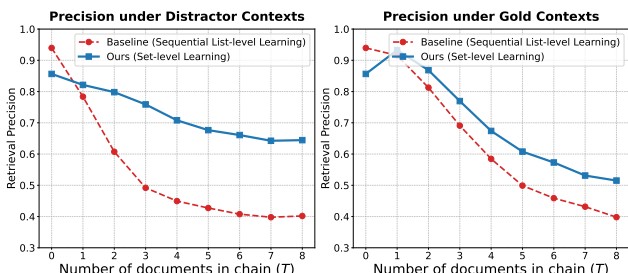

*Figure 1.* Analysis of sequential retrieval stability on the MuSiQue dataset under two controlled prefix settings. **Left (Distractor-only contexts):** the retrieved prefix consists entirely of irrelevant (distractor) passages. **Right (Partially-gold contexts):** for an $H$-hop question, the prefix is composed of only gold passages when $T < H$; once $T \geq H$, we include $H-1$ gold passages and fill the remaining positions with distractors, leaving exactly one unseen gold passage as the retrieval target. We compare a sequential list-level retriever trained with BCE (Baseline, Zhang et al.) against the *same retriever fine-tuned solely with our set-level compatibility learning objective* in section 3.1 (Ours). Importantly, both models share identical architectures and inference procedures. **Retrieval Precision** measures the success rate of retrieving the target gold passage given a prefix of length $T$. While the baseline exhibits a sharp degradation as the retrieval chain grows, the set-level objective substantially improves robustness to longer and noisier prefixes.

Recent work such as Lee et al. (2025) explicitly highlights this limitation and proposes retrieving sets of passages rather than individual passages. However, their approach relies on LLM-based set selection, resulting in substantial computational and memory overhead that limits its practicality in large-scale or latency-sensitive settings.

Although Chen et al. (2025) does not explicitly discuss the limitations of single-passage retrieval, it also aims to retrieve multiple objects jointly by leveraging structured relationships across heterogeneous sources. However, its retrieval pipeline depends on both an LLM and a MIP solver, incurring high computational cost.

Although Zhang et al. (2024) does not explicitly frame its motivation around set retrieval, its cross-encoder retriever can be directly used for multi-hop QA by iteratively constructing a retrieval chain using beam search. This procedure enables conditioning on previously retrieved evidence without relying on LLM-based passage selection. However, the training signal remains *local*: the pointwise binary classifi-

cation (BCE) objective supervises whether each candidate is a correct next passage given the current prefix, rather than directly optimizing the global compatibility between the query and the *entire* evidence set. In addition, the model is typically trained under a fixed-hop regime, while inference may involve longer prefixes and partially corrupted contexts due to error accumulation. As a result, the retriever is not explicitly encouraged to robustly evaluate query–set interactions under variable-length and noisy retrieval chains.

This limitation is illustrated in Figure 1. Under distractor-only prefixes (left), retrieval precision degrades rapidly as the chain grows, reflecting sensitivity to noise accumulation. Under partially-gold prefixes (right), the baseline remains relatively strong when the prefix aligns with the required hop structure, but its precision still declines as additional distractors are appended beyond the necessary evidence, indicating limited robustness to variable-length and increasingly noisy contexts.

**Our Approach.** Motivated by the above limitations, we address multi-hop retrieval from a set-level perspective by explicitly modeling query–set compatibility. Our approach departs from prefix-conditioned local supervision and instead operates directly on passage sets. Our contributions are summarized as follows:

- **Reframing multi-hop retrieval as a query–set scoring problem.** We formulate multi-hop retrieval as the problem of directly scoring the compatibility between a query and a *set of passages*, rather than as a sequence of local passage selection decisions. This perspective motivates learning objectives that operate at the set level, enabling explicit modeling of query–set interactions under variable-length retrieval chains.

- **A set-level compatibility learning objective for robust multi-hop retrieval.** We introduce a margin-based set-level learning objective that trains the retriever to distinguish compatible and incompatible passage sets. By explicitly incorporating noisy and partially corrupted retrieval prefixes during training, this objective substantially improves retrieval stability as chain length grows, even when applied as a fine-tuning step to existing retrievers.

- **An ultra-lightweight architecture for practical set-level retrieval.** Scoring passage sets naïvely is computationally infeasible, as a corpus of $N$ candidate passages induces $2^N - 1$ possible non-empty subsets. To make set-level scoring tractable despite this combinatorial explosion, we design `ParaSet` **(Parallel-Set scorer)**, an extremely lightweight single-layer attention-based model that operates on query and passage embeddings. This architecture supports efficient

batched scoring and integration with beam search, enabling fast exploration of promising passage sets.

## 2. Related Work

### 2.1. Multi-Passage Retrieval for Multi-hop QA

Early retrieval-based approaches for multi-hop question answering retrieve multiple passages independently, typically by selecting the top-$k$ most relevant passages for a given query (Karpukhin et al., 2020; Izacard & Grave, 2021; Lewis et al., 2020). Dense retrievers and cross-encoder models have been widely adopted in this setting, often followed by a downstream reader that performs multi-hop reasoning over the retrieved evidence (Min et al., 2019). While effective for aggregating information from multiple sources, these methods assume independent relevance among passages and do not explicitly model interactions or dependencies across retrieved evidence. As a result, they are limited in scenarios where the usefulness of a passage depends on its compatibility with other passages.

### 2.2. Sequential Multi-hop Retrieval

To address the limitations of independent retrieval, several studies perform multi-hop retrieval in an iterative manner, repeatedly retrieving passages conditioned on intermediate context (Xiong et al., 2020; Yao et al., 2022; Press et al., 2023; Trivedi et al., 2023; Yue et al., 2024). These approaches often reformulate the query or reasoning state at each step, enabling agentic exploration of the evidence space. In a different vein, Zhang et al. (2024) propose a cross-encoder-based retriever that predicts the next passage conditioned on the query and the previously retrieved passages, enabling sequential multi-hop retrieval without relying on large language models. More broadly, these methods formulate multi-hop retrieval as a sequence of local decisions, where candidate passages are scored step by step based on the evolving context. While this formulation allows retrievers to condition on previously retrieved passages, it remains inherently sequential and does not explicitly model the compatibility of the retrieved set as a whole.

### 2.3. Set-level Retrieval

Another line of work directly reasons over sets of passages via global selection. Recent work such as Lee et al. (2025) explicitly formulates retrieval as a set selection problem and leverages large language models to jointly reason over candidate passages. Similarly, Chen et al. (2025) retrieves multiple objects jointly by modeling structured relationships across heterogeneous sources, relying on a combination of LLMs and optimization solvers. Although these approaches capture set-level interactions, their reliance on LLM-based reasoning or complex inference pipelines incurs

substantial computational and memory overhead. In contrast, our approach directly scores candidate passage sets with a lightweight neural model, enabling efficient and scalable set-level retrieval without external solvers or LLM-based reasoning.

## 3. ParaSet: Parallel-Set Scorer

We propose `ParaSet`, a retrieval model designed to be computationally efficient while explicitly modeling the *semantic compatibility* of a set of passages. The model consists of two components: (1) a set-level compatibility learning objective, and (2) a lightweight single-layer self-attention architecture.

### 3.1. Set-level Compatibility Learning

To move beyond passage-level supervision and explicitly teach the model to reason over *sets of passages*, we introduce a set-level compatibility learning objective. Let the gold evidence set for a query $q$ be

$$\mathcal{G}_q = \{d_1^+, \ldots, d_k^+\}.$$

Directly enumerating and supervising all possible subsets of retrieved passages is practically infeasible due to the exponential number of combinations. Instead, inspired by the energy-based formulation of Song et al. (2025), we view set-level retrieval as a problem of learning relative scores over multiple passage sets. We construct informative negative sets with varying degrees of compatibility with $\mathcal{G}_q$, and train the model to assign higher score to the gold set than to incompatible alternatives. This encourages the model to shape an implicit energy surface over the combinatorial space of passage sets, without requiring exhaustive enumeration. The procedures used to construct negative sets are described in detail below.

**Negative set construction via structured perturbations.** We generate structured negative sets by perturbing the gold set $\mathcal{G}_q$ along three axes. For $d^- \notin \mathcal{G}_q$,

$$\textbf{Addition:} \quad N = \mathcal{G}_q \cup \{d^-\},$$
$$\textbf{Elimination:} \quad N \subset \mathcal{G}_q, \quad \text{where } N \neq \varnothing,$$
$$\textbf{Interchange:} \quad N = (\mathcal{G}_q \cup \{d^-\}) \setminus \{d_i^+\}.$$

These perturbations produce negative sets with systematically degraded compatibility, ranging from partially correct to severely corrupted combinations.

**Negative set construction via in-batch sampling.** In addition to structured perturbations, we incorporate *in-batch negative sets*. For a given query $q$, we treat gold evidence sets $\mathcal{G}_{q'}$ associated with other queries $q' \neq q$ within the same mini-batch as additional negatives. These in-batch negatives are often semantically plausible at the passage level but

incompatible with the current query. Furthermore, we construct additional negatives by randomly sampling subsets from all passages appearing in the same mini-batch. Together, these strategies provide a strong and diverse source of negative supervision.

**Set-level Compatibility Learning** The structured and in-batch negative sets described above exhibit varying degrees of compatibility with the gold evidence set. Since the notion of compatibility between passage sets depends on how set quality is defined, candidate sets can be ordered according to a pre-defined compatibility function. In our experiments, we rank sets lexicographically by prioritizing recall first and precision second for the multi-hop question answering. To provide fine-grained supervision, we apply a margin-based ranking loss to ordered pairs of candidate sets. Let $S(q, \cdot)$ denote the score assigned to a candidate passage set given a query $q$, and define the margin-based loss for an ordered pair $(H, L)$ (with $H \succ L$) as

$$\ell(H, L) = \max\left(0, \ \gamma - S(q, H) + S(q, L)\right),$$

where $\gamma$ is a margin hyperparameter. In our experiments, we define set quality based on its agreement with the gold evidence set for the query, and rank candidate sets lexicographically by prioritizing *recall* with respect to the gold set first, followed by *precision*.

In practice, we construct a candidate pool $\mathcal{C}$ for each query, which includes the gold evidence set, structured negative sets generated via set-level perturbations (e.g., addition, elimination, and interchange), as well as in-batch negative sets sampled from other training instances.

We enforce a coarse-grained constraint requiring the gold set to outrank all other candidate sets:

$$\mathcal{L}_{\text{pos-all}} = \sum_{L \in \mathcal{C} \setminus \{\mathcal{G}_q\}} \ell(\mathcal{G}_q, L).$$

This term ensures that the model assigns the highest compatibility score to the gold evidence set.

To provide finer-grained supervision, we impose additional ranking constraints between sets with *adjacent compatibility levels*, as induced by a predefined ordering function. Let $\mathcal{P}_{\text{adj}}$ denote the set of ordered pairs $(H, L)$ such that $H$ is ranked immediately above $L$. We define

$$\mathcal{L}_{\text{adj}} = \sum_{(H, L) \in \mathcal{P}_{\text{adj}}} \ell(H, L).$$

The final objective is given by

$$\mathcal{L}_{\text{hard}} = \mathcal{L}_{\text{pos-all}} + \lambda \, \mathcal{L}_{\text{adj}}, \tag{1}$$

where $\lambda$ controls the relative weight of the fine-grained adjacent-ranking constraints.

### 3.2. Single-layer Self-Attention Architecture

**Architecture** The `ParaSet` scorer adopts a lightweight *single-layer self-attention architecture* built on top of bi-encoder representations. Given a query embedding $q$ and candidate passage embeddings $\{p_1, \ldots, p_n\}$ obtained from a bi-encoder, we construct a short input sequence consisting of a learnable `[CLS]` token, the query embedding, and the passage embeddings in the set. We then apply a single multi-head self-attention layer over this sequence to model interactions between the query and the set of passages.

Unlike deep cross-encoders, the `ParaSet` scorer employs only a single self-attention layer, which allows it to capture global interactions between the query and passages in a set while maintaining low computational overhead. This design is particularly well suited for multi-hop retrieval, where a large number of candidate passage sets must be evaluated efficiently. In particular, the efficiency of the set scorer stems from three key design choices.

First, because the scorer consists of only a single attention layer and produces the final score solely from the `[CLS]` representation, meaningful attention outputs are required only for the `[CLS]` token. Although a standard self-attention layer is applied, representations of other tokens are not propagated to subsequent layers, which substantially limits the effective computation.

Second, passage embeddings are produced by a query-independent bi-encoder, allowing their key and value projections to be precomputed and cached. As a result, at inference time the set scorer incurs only minimal query-dependent computation on top of cached passage representations.

Third, the shallow and lightweight nature of the architecture enables efficient parallel evaluation of a large number of candidate passage sets. This makes it practical to explore the combinatorial space of passage sets in multi-hop retrieval.

Together, these three properties allow the set scorer to introduce only minimal additional computation on top of standard dense retrieval, while still enabling explicit set-level modeling through attention-based aggregation.

**Beam-search set retrieval.** Our goal is to approximately solve

$$S^\star = \underset{S \subseteq [n], S \neq \varnothing}{\arg\max} \ s(q, S),$$

which is intractable to enumerate due to the exponential number of subsets. We therefore use beam search with batched set scoring. The beam is initialized with the top-$B$ singleton sets, and at each iteration, each beam subset is expanded by adding one new passage. All generated subsets are scored in parallel, and the top-$B$ subsets are retained for the next iteration. The final output is chosen as the highest-scoring subset among *all sets evaluated during the search*,

rather than being restricted to the final beam.

The efficiency of this search procedure relies on the lightweight nature of `ParaSet`. The lightweight architecture allows many candidate sets associated with the same query to be evaluated simultaneously, which is critical for efficient beam search over the combinatorial set space. This makes parallel set-level scoring practical, in contrast to heavier cross-encoder or LLM-based retrievers.

Additional details of the beam-search set retrieval algorithm is provided in Appendix B.

## 4. Experiments

### 4.1. Experimental Setup

**Benchmarks.** We evaluate our method on three widely used multi-hop question answering benchmarks: HotpotQA (Yang et al., 2018), 2WikiMultihopQA (Ho et al., 2020), and MuSiQue (Trivedi et al., 2022). HotpotQA consists exclusively of 2-hop questions, while 2WikiMultihopQA and MuSiQue include questions requiring up to 4 hops. These datasets allow us to assess retrieval and end-task QA performance across a range of reasoning depths.

**Baselines.** We compare `ParaSet` against a set of strong retrieval baselines that represent different modeling paradigms. **Bi** is a dense bi-encoder retriever based on Contriever (Izacard et al., 2022), which independently encodes queries and passages and retrieves top-$k$ passages using vector similarity. **CE** is a cross-encoder retriever based on `ms-marco-electra-base` (Hugging Face, 2021), which jointly encodes each query–passage pair to produce relevance scores. **ListCE** is the sequential list-level retriever proposed by Zhang et al. (2024), implemented with a `deberta-v3-base` backbone and trained using a pointwise binary classification objective. This model iteratively expands a retrieval chain via beam search, and selects the final output as the highest-scoring passage set among all candidates evaluated during search, following the same inference protocol as `ParaSet`.

To isolate the effect of set-level compatibility learning from architectural factors, we additionally consider **SetCE**, a variant of `ListCE` fine-tuned using our set-level compatibility learning objective (Section 3.1), while keeping the model architecture and inference procedure identical. This comparison allows us to evaluate the impact of the training objective while controlling for architectural and inference differences.

**Hybrid reranking with ParaSet.** Beam-search-based retrieval methods such as `ListCE` and `SetCE` score candidate passage sets by repeatedly applying a cross-encoder to partially constructed passage sets during beam search. Directly applying these models to evaluate all candidates

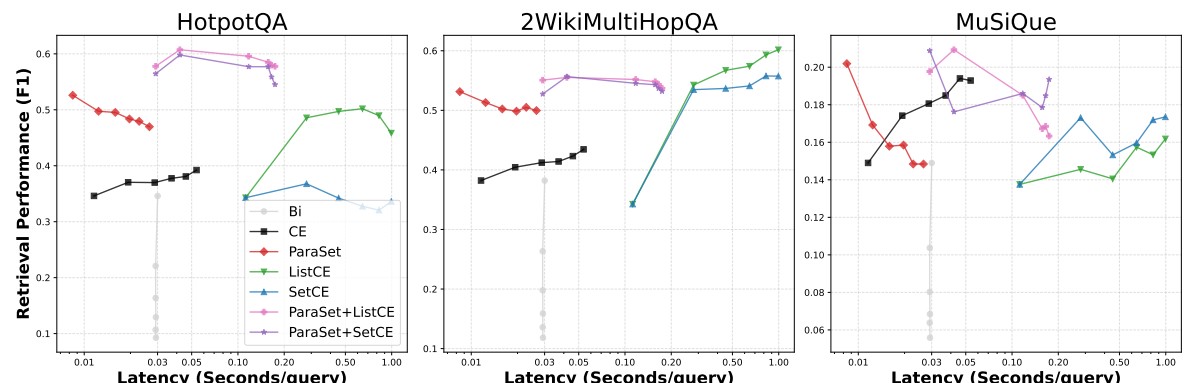

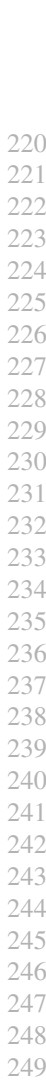

*Figure 2.* **Trade-off between retrieval performance and inference latency** across three multi-hop question answering datasets: HotpotQA, 2WikiMultiHopQA, and MuSiQue. The $x$-axis represents the inference latency per query (seconds per query) of the reranking stage, plotted on a logarithmic scale, while the $y$-axis shows retrieval performance measured by F1 score. Each point corresponds to a different number of candidate documents $K \in \{5, 10, \dots, 30\}$ retrieved by the bi-encoder. Retrieval F1 is computed by comparing the final selected passage set against the gold supporting passages. Lower values on the $x$-axis indicate faster inference, making the **upper-left region of the plot** correspond to more favorable performance–efficiency trade-offs.

generated during beam search can incur a large number of forward passes, resulting in high inference cost. To mitigate this issue, we consider hybrid reranking pipelines that combine `ParaSet` with these beam-search-based models, using `ParaSet` to efficiently filter and prioritize candidate sets before applying heavier scoring functions. In these hybrid settings, `ParaSet` is first used as a lightweight set scorer to efficiently explore the combinatorial set space and identify a small pool of top-$S$ candidate sets. These candidates are then re-scored by a more expressive cross-encoder-based model, requiring exactly one forward pass per candidate set. We evaluate two variants: **ParaSet + ListCE** and **ParaSet + SetCE**, which differ only in the choice of the final reranker. This hybrid design uses `ParaSet` to efficiently explore and prune candidate passage sets, while reserving `ListCE` or `SetCE` for fine-grained compatibility evaluation, thereby reducing inference cost while maintaining competitive performance.

### 4.2. Retrieval Effectiveness–Efficiency Trade-off

Figure 2 illustrates the trade-off between retrieval performance (F1) and inference speed across three multi-hop QA datasets. For this analysis, we randomly sample 100 queries from the dataset and report the average retrieval speed and retrieval F1. We vary the number of retrieved documents $K \in \{5, 10, 15, \dots, 30\}$ from the bi-encoder, connecting results with the same $K$ using a consistent color. Except for the bi-encoder, all reported inference times exclude the cost of query embedding and FAISS-based dense retrieval. That is, for all non-bi-encoder settings, we measure only the additional computation incurred by the corresponding retrieval or reranking module applied on top of a fixed set of candidate passages. This allows us to focus on the relative computational overhead introduced by different retrieval and

*Table 1.* Number of retrieved passages for each model on 2Wiki-MultiHopQA. We report the average number of passages selected after reranking for different candidate sizes $K$. The average number of gold supporting passages is 2.5.

| Model | $K = 20$ | $K = 30$ |
|---|---|---|
| CE | 5.0 | 5.0 |
| ListCE | 2.9 | 3.0 |
| SetCE | 3.3 | 3.5 |
| ParaSet | 2.4 | 2.5 |
| ParaSet + ListCE | 2.6 | 2.8 |
| ParaSet + SetCE | 2.6 | 2.7 |

reranking strategies beyond dense retrieval. We note that bi-encoder and cross-encoder baselines differ from other retrieval settings in that the number of documents used for evaluation must be fixed in advance. For the bi-encoder baseline, we compute retrieval F1 using all $K$ retrieved passages. In contrast, for cross-encoder-based methods, we re-rank the $K$ candidates and evaluate retrieval F1 using the top-5 passages.

Several observations can be drawn from Figure 2. First, `ParaSet` tends to achieve higher retrieval F1 than the cross-encoder baseline while exhibiting faster inference speed. This indicates that lightweight set-level scoring can provide stronger retrieval effectiveness than passage-level cross-encoders without increasing latency.

Second, combining `ParaSet` with additional reranking modules (`ParaSet+ListCE` and `ParaSet+SetCE`) yields a particularly strong performance–speed trade-off. These pipelines substantially improve retrieval F1, incurring additional inference time compared to standalone `ParaSet`. Nevertheless, the added computational cost remains moderate, indicating that selective reranking on top of efficient

*Table 2.* End-task QA performance grouped by question hop count. We report EM / F1 for each hop, along with the macro-average across available hops for each dataset.

| Dataset | Retriever | 2-hop | | 3-hop | | 4-hop | | Macro Avg | |
|---|---|---|---|---|---|---|---|---|---|
| | | EM | F1 | EM | F1 | EM | F1 | EM | F1 |
| HotpotQA | Bi | 0.3500 | 0.4590 | - | - | - | - | 0.3500 | 0.4590 |
| | Bi + CE | 0.3903 | 0.5059 | - | - | - | - | 0.3903 | 0.5059 |
| | Bi + ListCE | 0.3865 | 0.4973 | - | - | - | - | 0.3865 | 0.4973 |
| | Bi + SetCE | 0.3910 | 0.5058 | - | - | - | - | 0.3910 | 0.5058 |
| | Bi + ParaSet | 0.3661 | 0.4788 | - | - | - | - | 0.3661 | 0.4788 |
| | Bi + ParaSet + ListCE | 0.3758 | 0.4903 | - | - | - | - | 0.3758 | 0.4903 |
| | Bi + ParaSet + SetCE | **0.4041** | **0.5222** | - | - | - | - | **0.4041** | **0.5222** |
| 2Wiki | Bi | 0.2364 | 0.2748 | - | - | 0.2008 | 0.2113 | 0.2186 | 0.2430 |
| | Bi + CE | **0.3004** | **0.3429** | - | - | 0.2672 | 0.2751 | 0.2838 | 0.3090 |
| | Bi + ListCE | 0.2884 | 0.3318 | - | - | 0.2132 | 0.2219 | 0.2508 | 0.2769 |
| | Bi + SetCE | 0.2988 | 0.3425 | - | - | 0.1827 | 0.1911 | 0.2407 | 0.2668 |
| | Bi + ParaSet | 0.2584 | 0.3029 | - | - | 0.2679 | 0.2775 | 0.2631 | 0.2902 |
| | Bi + ParaSet + ListCE | 0.2776 | 0.3245 | - | - | 0.2817 | 0.2898 | 0.2796 | 0.3071 |
| | Bi + ParaSet + SetCE | 0.2746 | 0.3206 | - | - | **0.2988** | **0.3425** | **0.2867** | **0.3315** |
| MuSiQue | Bi | 0.1108 | 0.1810 | **0.0980** | **0.1768** | 0.0446 | 0.1088 | 0.0845 | 0.1556 |
| | Bi + CE | **0.1188** | 0.1878 | 0.0846 | 0.1658 | 0.0525 | 0.1229 | 0.0853 | 0.1588 |
| | Bi + ListCE | 0.1075 | 0.1817 | 0.0926 | 0.1733 | 0.0420 | 0.1078 | 0.0807 | 0.1543 |
| | Bi + SetCE | **0.1188** | **0.1881** | 0.0953 | 0.1728 | 0.0551 | **0.1266** | **0.0898** | **0.1625** |
| | Bi + ParaSet | 0.0865 | 0.1520 | 0.0604 | 0.1347 | 0.0394 | 0.1015 | 0.0621 | 0.1294 |
| | Bi + ParaSet + ListCE | 0.0986 | 0.1616 | 0.0725 | 0.1484 | 0.0420 | 0.1024 | 0.0710 | 0.1375 |
| | Bi + ParaSet + SetCE | 0.1035 | 0.1708 | 0.0577 | 0.1342 | **0.0577** | 0.1167 | 0.0730 | 0.1406 |

set-level retrieval can be an effective way to improve retrieval quality.

A notable observation is that, unlike the other datasets, MuSiQue shows a more decline in performance for `ParaSet`-based models as the candidate size $K$ increases. One possible contributing factor is the relatively lower retrieval quality of the bi-encoder on MuSiQue. While the bi-encoder attains retrieval F1 scores of approximately 0.3–0.4 on HotpotQA and 2WikiMultiHopQA, its retrieval F1 on MuSiQue is substantially lower. This indicates that the behavior of `ParaSet`-based methods on MuSiQue may be influenced by the quality of the initial candidate passages available for set construction.

Another potential explanation relates to structural differences in how evidence passages are connected across datasets. In HotpotQA and 2WikiMultiHopQA, the underlying reasoning graphs linking relevant passages tend to be relatively simple and shallow, whereas MuSiQue often requires more complex and less explicit compositional reasoning across multiple passages. In such settings, alternative training strategies—such as curriculum learning approaches that progressively move from simpler reasoning structures to more complex ones—may be beneficial. However, we do not further investigate these factors in this work and leave a detailed analysis of dataset-specific structural effects to future research.

Table 1 reports the average number of passages selected by

each retrieval or reranking model on 2WikiMultiHopQA for different candidate sizes $K$. The cross-encoder (CE) baseline consistently selects five passages, reflecting its fixed top-5 evaluation protocol, independent of the candidate pool size. In contrast, ParaSet-based methods adaptively select a variable number of passages, resulting in set sizes that are generally closer to the average number of gold supporting passages (2.5). This behavior suggests that set-level retrieval encourages more selective evidence aggregation, avoiding the inclusion of unnecessary passages when fewer documents suffice.

Finally, while higher retrieval F1 generally reflects stronger retrieval effectiveness, it does not fully determine downstream QA performance in multi-hop settings. This phenomenon will be discussed in detail in Section 4.3. Further details regarding the experimental settings and retrieval speed measurements can be found in Appendix D.

### 4.3. End-to-End Performance

The end-task QA performance results are summarized in Table 2, grouped by the number of required reasoning hops. For end-to-end evaluation, the retrieved passage sets produced by different retrieval methods are passed, together with a fixed set of few-shot exemplars, to a frozen API-based LLM (`gpt-4o-mini`, OpenAI) for answer generation. No task-specific fine-tuning is performed for LLM; thus, all performance differences arise solely from variations in the

upstream retrieval and reranking stages.

For the bi-encoder baseline, the top-5 retrieved passages are directly provided to the LLM. For CE, ListCE, and SetCE, we first retrieve 20 candidate passages using the bi-encoder and apply reranking on this fixed pool; in particular, CE selects a fixed top-5 subset for answer generation. Due to its lightweight design, ParaSet can explore a larger candidate space within a comparable time budget, and we therefore allow it to perform beam search over up to 50 candidate passages. For hybrid variants (ParaSet + ListCE and ParaSet + SetCE), ParaSet first retrieves candidate passage sets from the same pool of 50 passages and selects the top-10 highest-scoring sets, which are then passed to the subsequent reranker(ListCE/SetCE) and ultimately to the LLM. We highlight the following observations.

First, the advantage of set-level retrieval models becomes more evident as the hop count increases. In particular, models incorporating set-level compatibility learning (i.e., SetCE and ParaSet + SetCE) tend to achieve stronger answer generation performance on higher-hop questions. In low-hop scenarios, their performance is often comparable to, or slightly behind, simpler alternatives; however, as the number of required hops grows, set-level modeling becomes increasingly beneficial. This suggests that explicitly learning scores over query–set pairs is more effective when retrieving larger and more complex evidence sets, where holistic assessment of passage compatibility plays a larger role.

Second, when comparing ListCE with SetCE, as well as ParaSet + ListCE with ParaSet + SetCE, we observe that SetCE outperforms ListCE in most settings. We attribute this behavior to the fact that SetCE is trained to reason over the retrieved set as a whole, rather than optimizing a purely sequential ranking objective. As illustrated in Figure 1 (ListCE v.s. SetCE), this holistic training objective improves robustness under diverse and non-ideal retrieval conditions, which frequently arise in multi-hop and agentic retrieval pipelines.

Unfortunately, ParaSet alone generally does not outperform strong baseline retrievers in terms of end-task QA performance. However, this behavior aligns with its intended role in the retrieval pipeline. Analogous to the standard document-level retrieve-and-rerank paradigm—where a bi-encoder first retrieves candidate documents from a large corpus and a cross-encoder subsequently performs fine-grained reranking—ParaSet and SetCE operate as a lightweight retrieve-and-rerank pair at the level of passage sets.

Specifically, ParaSet serves as an efficient set-level retriever that rapidly generates and filters candidate passage sets, while SetCE performs more expressive set-level compatibility scoring over these candidates. When combined as ParaSet + SetCE, this division of labor enables effective

*Table 3.* Comparison between multi-step (agentic) and single-step retrieval settings on the MuSiQue dataset. We report EM / F1 for each hop category and the macro average.

| Hop | Retrieval Setting | Model | EM | F1 |
|-----|------------------|-------|----|----|
| 2-hop | Multi-step | Bi | 0.1172 | 0.2021 |
| | | Bi + CE | **0.1334** | **0.2175** |
| | Single-step | Bi + SetCE | 0.1188 | 0.1881 |
| | | Bi + ParaSet + SetCE | 0.1035 | 0.1708 |
| 3-hop | Multi-step | Bi | 0.0819 | 0.1725 |
| | | Bi + CE | 0.0805 | 0.1685 |
| | Single-step | Bi + SetCE | **0.0953** | **0.1728** |
| | | Bi + ParaSet + SetCE | 0.0577 | 0.1342 |
| 4-hop | Multi-step | Bi | 0.0551 | **0.1310** |
| | | Bi + CE | 0.0472 | 0.1281 |
| | Single-step | Bi + SetCE | 0.0551 | 0.1266 |
| | | Bi + ParaSet + SetCE | **0.0577** | 0.1167 |
| Macro Avg | Multi-step | Bi | 0.0847 | 0.1685 |
| | | Bi + CE | 0.0871 | **0.1714** |
| | Single-step | Bi + SetCE | **0.0898** | 0.1625 |
| | | Bi + ParaSet + SetCE | 0.0730 | 0.1406 |

set-level evaluation without incurring the high computational cost of applying expensive scorers over the full search space.

## 4.4. Comparison with Agentic Retrieval Pipelines

We further compare our retrieve paradigm against agentic retrieval pipelines that iteratively interleave sub-query generation, document retrieval, and intermediate answer generation. Representative examples of such approaches include IRCoT-style pipelines (Trivedi et al., 2023), where an LLM regenerates the original question to a new query and performs retrieval multiple times conditioned on intermediate reasoning states.

In this setting, the retrieval component plays a critical role, as retrieval errors may compound across iterations. To assess the effectiveness of our set-level retrieval methods under this more challenging regime, we compare against agentic pipelines equipped with (i) a bi-encoder retriever and (ii) a bi-encoder followed by a cross-encoder reranker. At each iteration the bi-encoder retrieves three passages, while the Bi+CE variant reranks 20 candidates and selects the top-3 passages to be provided to the LLM. Experiment is conducted on the MuSiQue dataset, which contains a diverse mixture of 2-hop to 4-hop questions.

Following the same evaluation protocol as in Section 4.3, we directly provide the retrieved passages to the LLM for answer generation without iterative query refinement. That is, unlike agentic pipelines, our methods perform retrieval only once and do not involve intermediate sub-query generation or repeated retrieval steps.

Table 3 summarizes the end-task QA performance. Despite relying on a single retrieval step without iterative LLM-

*Table 4.* Retrieval F1 and relative inference speed of `ParaSet` on MuSiQue with different beam widths. Inference speed is normalized by the beam width of 1.

| Beam Width | Retrieval F1 | Relative Speed |
|:----------:|:------------:|:--------------:|
| 1 | 0.2347 | 1.000 |
| 2 | **0.2375** | 0.677 |
| 3 | 0.2354 | 0.473 |
| 4 | 0.2350 | 0.401 |
| 5 | 0.2353 | 0.341 |

driven query refinement, our proposed methods achieve performance that is competitive with, and in some cases superior to, agentic retrieval pipelines. In particular, `SetCE` matches or outperforms agentic baselines across most hop settings, including higher-hop questions where iterative retrieval is typically assumed to be advantageous.

Notably, these results are achieved with substantially lower inference cost. While agentic pipelines require multiple LLM calls for sub-query generation, retrieval, and intermediate reasoning at each hop, our approach performs retrieval once and directly supplies a passage set to the LLM. In our IRCoT-style implementation, both the bi-encoder and Bi+CE agentic variants perform an average of 3.7 retrieval iterations per question, each involving additional LLM calls and retrieval operations. By contrast, our retrieve-at-once paradigm eliminates this iterative interaction, relying instead on explicit set-level compatibility modeling to identify globally coherent evidence in a single step. This suggests that set-level compatibility learning can partially substitute for iterative reasoning in agentic pipelines.

## 5. Additional Experiments

### 5.1. Beam Width Analysis for ParaSet

We analyze the effect of beam width during `ParaSet` inference to understand the trade-off between retrieval quality and computational efficiency. In this experiment, we evaluate `ParaSet` on the MuSiQue dataset while varying the beam width used during candidate set generation.

Following our standard retrieval pipeline, we first retrieve the top-20 documents using a bi-encoder. `ParaSet` is then applied to these candidates using beam search with different beam widths. For each setting, we report retrieval F1 and relative inference speed, normalized by the beam width of 1. Table 4 summarizes the results. We observe that increasing the beam width from 1 to 2 yields a noticeable improvement in retrieval F1, while incurring a moderate computational overhead. However, further increasing the beam width beyond 2 does not lead to additional performance gains, and instead results in diminishing returns with substantially increased inference cost.

### 5.2. Effect of Adjacent-Ranking Weight.

The weight $\lambda$ of the adjacent-ranking term controls the balance between *coverage* and *parsimony* in set-level retrieval. Empirically, we observe that increasing $\lambda$ can bias the model toward either more compact or more inclusive evidence sets, depending on the dataset. This indicates that there is no universal optimal value of $\lambda$ across benchmarks, and its effect is closely tied to dataset-specific evidence characteristics. A detailed quantitative analysis is provided in Appendix E.

## 6. Conclusion

We revisited multi-hop retrieval from a set-level perspective and argued that effective evidence selection requires modeling *query–set compatibility*, rather than independently scoring passages or relying solely on sequential, locally supervised decisions. To this end, we introduced `ParaSet` with two complementary components: (i) a set-level compatibility learning objective that directly distinguishes compatible and incompatible passage sets and improves robustness under variable-length and noisy retrieval chains, and (ii) `ParaSet`, an lightweight single-layer self-attention scorer that enables efficient exploration of the combinatorial passage-set space via parallel batched scoring.

Across HotpotQA, 2WikiMultiHopQA, and MuSiQue datasets, our experiments demonstrate that set-level compatibility learning becomes increasingly beneficial as the required reasoning hop count grows. While `ParaSet` alone is not always the strongest end-to-end retriever in terms of absolute QA performance, it consistently offers a favorable effectiveness–efficiency trade-off and serves as a practical intermediate candidate selector. In particular, combining `ParaSet` with stronger set-aware rerankers such as `SetCE` achieves competitive end-task QA performance while substantially reducing inference cost.

Moreover, we show that single-step retrieval can remain competitive even when compared against agentic multi-step retrieval pipelines such as IRCoT, which rely on multiple rounds of LLM-driven sub-query generation and retrieval. These results suggest that explicit set-level compatibility modeling can partially substitute for iterative reasoning in agentic pipelines, enabling efficient identification of globally coherent evidence sets without incurring the overhead of multi-step interaction.

Our findings also highlight several limitations and directions for future work. First, the benefits of set-level retrieval can be constrained by the quality of the initial bi-encoder candidate pool, as suggested by the different behavior observed on MuSiQue. Second, hyperparameters that govern fine-grained set supervision (e.g., the adjacent-ranking weight) exhibit dataset-dependent effects, motivating adaptive strategies for tuning and learning such objectives.

## 7. Impact Statement

This paper presents research whose primary goal is to advance the field of machine learning, with a particular focus on retrieval methods for multi-hop question answering and retrieval-augmented reasoning. The proposed techniques improve the robustness and efficiency of evidence retrieval by modeling compatibility over sets of documents, which may contribute to more reliable downstream reasoning systems.

We do not foresee any specific negative societal consequences that are unique to the techniques proposed in this work. As such, we believe that the ethical considerations and broader impacts are consistent with well-established practices in machine learning research.

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

## A. Limitations of Pointwise Training under Variable-Length Retrieval

**Failure Modes of Sequential Pointwise Training.**  We analyze the failure modes of the sequential list-level retriever proposed by Zhang et al. (2024), which is trained using a pointwise binary classification (BCE) objective. In this framework, the model scores each candidate passage independently conditioned on the current retrieval prefix, and the retrieval chain is expanded sequentially via beam search.

A key limitation of this training strategy is that the model is never explicitly optimized to evaluate the compatibility of an entire set of retrieved passages. Instead, supervision is provided only for local, step-wise decisions, implicitly assuming that the retrieval prefix remains within the distribution observed during training.

This assumption breaks down in practice for two reasons. First, during inference, the retrieval chain can grow beyond the fixed hop count used during training, resulting in prefixes whose length and composition differ from those seen during optimization. Second, retrieval errors can accumulate over steps, introducing irrelevant passages into the prefix and further amplifying distribution shift.

As demonstrated in Figure 1, these effects lead to a sharp degradation in retrieval precision as the chain length increases, even under controlled partially-gold prefix settings. This behavior suggests that pointwise training fails to equip the model with the ability to robustly score query–set interactions under variable-length and partially corrupted retrieval contexts.

## B. Details of the `ParaSet` Scorer and Beam-Search Inference

This appendix provides implementation-level details of the `ParaSet` scorer and its beam-search inference procedure.

### B.1. Notation

Given an input query, a bi-encoder produces a query embedding ($q \in \mathbb{R}^d$) and candidate passage embeddings $\{p_1, \ldots, p_n\}$ where $p_i \in \mathbb{R}^d$. We write a candidate set by its index subset $S \subseteq [n]$ and denote the corresponding passage embeddings by $\{p_i\}_{i \in S}$. In the attention layer, we use ($Q$) for the attention query vector, and ($K, V$) for key/value vectors:

$$Q = W_Q q, \qquad K_i = W_K p_i, \qquad V_i = W_V p_i.$$

We use a beam width $B$.

### B.2. Forward Computation of the Set Score

To score a set $S$, we construct an input sequence consisting of a learnable `[CLS]` token, the query embedding ($q$), and the passage embeddings in the set:

$$x(S) = (\texttt{[CLS]}, q, \{p_i\}_{i \in S}).$$

We apply a single multi-head self-attention layer followed by a lightweight feed-forward block. The final set score is obtained from the `[CLS]` output:

$$s(q, S) = w^\top h_{\texttt{[CLS]}}(S),$$

where $h_{\texttt{[CLS]}}(S)$ is the `[CLS]` representation after the single attention layer (and optional FFN).

### B.3. Beam-Search Set Retrieval

Our objective is to approximately solve

$$S^\star = \arg \max_{S \subseteq [n],\, S \neq \varnothing} s(q, S),$$

where $n$ is the number of candidate passages. Since there are $2^n - 1$ non-empty subsets, exhaustive enumeration is infeasible except for very small $n$. We therefore employ a *depth-limited beam search* with batched set scoring.

In practice, we restrict the maximum size of candidate sets during inference to a predefined limit $T_{\max}$. This constraint reflects the observation that multi-hop question answering typically requires only a small number of supporting passages, and it prevents unnecessary expansion into overly large and noisy sets.

**Initialization and expansion.** We first score all singleton sets and retain the top-$B$ as the initial beam. At each iteration, every beam subset $S$ is expanded by adding new passages from the remaining pool. The expansion proceeds only if $|S| < T_{\max}$, ensuring that the beam search explores sets up to a bounded size.

**Expansion.** For step $t = 2, 3, \ldots, T_{\max}$, we expand each beam subset by adding new elements from the remaining pool. Let $a_{\max}$ denote the maximum number of newly added passages per expansion step (in our standard setting, $a_{\max} = 1$). For each $S \in \mathcal{B}_{t-1}$ with $|S| < T_{\max}$, we generate candidate sets

$$\mathcal{N}(S) = \{S \cup A : A \subseteq [n] \setminus S, \ 1 \le |A| \le a_{\max}\}.$$

All generated candidates are scored in batch, deduplicated, and the top-$B$ subsets are retained as the next beam $\mathcal{B}_t$.

---

**Algorithm 1** Depth-limited beam search for set retrieval with `ParaSet`

---

**Require:** query embedding $q$, candidate passages $\{p_1, \ldots, p_n\}$, beam width $B$, maximum set size $T_{\max}$
1: $\mathcal{S}_{\text{eval}} \leftarrow \emptyset$
2: $\mathcal{B} \leftarrow \text{TopB}(\{\{i\}\}_{i=1}^n)$
3: $\mathcal{S}_{\text{eval}} \leftarrow \mathcal{S}_{\text{eval}} \cup \mathcal{B}$
4: **while** true **do**
5:    $\mathcal{C} \leftarrow \emptyset$
6:    **for** each $S \in \mathcal{B}$ **do**
7:      **if** $|S| < T_{\max}$ **then**
8:        **for** each $j \in [n] \setminus S$ **do**
9:          $\mathcal{C} \leftarrow \mathcal{C} \cup \{S \cup \{j\}\}$
10:        **end for**
11:      **end if**
12:    **end for**
13:    **if** $\mathcal{C} = \emptyset$ **then**
14:      **break**
15:    **end if**
16:    Score all $S \in \mathcal{C}$ in batch (with memoization)
17:    $\mathcal{S}_{\text{eval}} \leftarrow \mathcal{S}_{\text{eval}} \cup \mathcal{C}$
18:    $\mathcal{B} \leftarrow \text{TopB}(\mathcal{C})$
19: **end while**
20: **return** $\text{Rank}(\mathcal{S}_{\text{eval}})$

---

## C. Training Details

This appendix provides additional implementation details for training `ParaSet` (ParaSet) and SetCE.

### C.1. ParaSet Architecture and Training

**Model Architecture.** `ParaSet` is implemented as a Transformer encoder designed for efficient set-level scoring. We use a single-layer Transformer encoder with a hidden dimension of $d_{\text{model}} = 768$ and 8 attention heads. The feed-forward network has an intermediate dimension of $d_{\text{ff}} = 2048$. Following standard practice, we use residual connections and layer normalization. The final set-level score is computed from the `[CLS]` representation.

**Training Configuration.** `ParaSet` is trained using the AdamW optimizer with a learning rate of $5 \times 10^{-5}$. We use a batch size of 8, where each mini-batch is constructed to support in-batch negative sampling. Models are trained for 200 epochs. Unless otherwise specified, all other hyperparameters follow standard Transformer defaults.

**Negative Sampling Strategy.** We employ two complementary sources of negative supervision: (1) structured set-level perturbations constructed from a bi-encoder candidate pool, and (2) in-batch negatives sampled exclusively from within each mini-batch.

*Table 5.* Inference latency (seconds per query) for different retrieval and reranking models on 2WikiMultiHopQA. All measurements are conducted with batch size 1 on a single NVIDIA RTX 3090 GPU.

| Model | $K = 10$ | $K = 20$ | $K = 30$ | $K = 40$ | $K = 50$ | $K = 60$ | $K = 70$ |
|---|---|---|---|---|---|---|---|
| Bi-Encoder | 0.0291 | 0.0292 | 0.0293 | 0.0292 | 0.0291 | 0.0291 | 0.0292 |
| CE | 0.0500 | 0.0677 | 0.0845 | 0.1050 | 0.1228 | 0.1401 | 0.1605 |
| ListCE / SetCE | 0.3138 | 0.6825 | 1.0349 | 1.4128 | 1.8032 | 2.2113 | 2.5862 |
| ParaSet | 0.0424 | 0.0512 | 0.0588 | 0.0640 | 0.0705 | 0.0776 | 0.0861 |
| ParaSet + (ListCE / SetCE) | 0.0740 | 0.1915 | 0.2085 | 0.2214 | 0.2390 | 0.2579 | 0.2720 |

**Structured perturbation negatives.** For each query, we first retrieve the top-20 candidate documents using a bi-encoder. Structured negative sets are then generated from this fixed candidate pool by applying set-level perturbations described in Section 3.1, including addition, elimination, and interchange operations. These negatives explicitly model partially corrupted or noisy passage sets that are close to the gold evidence set in structure.

**In-batch negatives.** In-batch negatives are constructed solely from other training instances within the same mini-batch, independent of the bi-encoder retrieval results for the current query. Specifically, we include:

- **In-batch positive negatives:** Gold evidence sets from other samples in the same mini-batch ($n = 4$).

- **In-batch random negatives:** Randomly sampled subsets constructed from passages belonging to other samples in the same mini-batch ($n = 4$).

These in-batch negatives are often semantically plausible at the passage level but incompatible with the current query, providing additional supervision that encourages query-specific set discrimination.

### C.2. SetCE Fine-tuning Details

SetCE is obtained by fine-tuning a pretrained ListCE model using the proposed set-level compatibility learning objective, while keeping the model architecture and inference procedure unchanged.

**Training Configuration.** Fine-tuning is performed using the AdamW optimizer with a weight decay of $10^{-2}$. We use a learning rate of $1 \times 10^{-5}$ and a batch size of 2 per NVIDIA A6000 GPU. Training is conducted for a single epoch, which we found sufficient to adapt the model to the set-level compatibility objective without overfitting.

## D. Experimental Setup and Efficiency Analysis

This appendix provides detailed implementation and evaluation settings for the retrieval effectiveness–efficiency analysis presented in Section 4.2. In particular, we describe the hardware and inference configurations used to measure retrieval latency, as well as additional quantitative results on retrieval speed across different retrieval and reranking pipelines.

**Hardware Configuration** All inference speed measurements were conducted on a single NVIDIA RTX 3090 GPU. To reflect realistic per-query latency in retrieval-augmented question answering scenarios, all models were evaluated with a batch size of 1, i.e., queries were processed sequentially. Inference latency was measured as wall-clock time from query input to the final selection of the retrieved passage set.

**Inference Configuration** For all beam-search-based methods, including `ParaSet`, `ListCE`, `SetCE`, and their hybrid variants, we fix the maximum retrieved set size to 9. That is, beam search is performed up to a depth of 9, corresponding to selecting at most 9 passages per retrieved set. The beam size is fixed to 2 for all experiments.

For hybrid reranking pipelines (`ParaSet + ListCE` and `ParaSet + SetCE`), `ParaSet` is first used to generate and score candidate passage sets. The top-10 highest-scoring sets are then passed to the subsequent reranking stage. As a result, the cross-encoder-based reranker performs exactly 10 forward passes per query, independent of the candidate size $K$ or beam search depth. This design explicitly bounds the computational cost of the reranking stage.

*Table 6.* Average size of the top-ranked set predicted by `ParaSet` under different $\lambda$ in Eq. 1.

| Dataset | $\lambda = 0.2$ | $\lambda = 0.5$ | $\lambda = 0.8$ |
|---|---|---|---|
| HotpotQA | 3.049 | 19.80 | 19.88 |
| 2WikiMultihopQA | 4.081 | 3.550 | 3.352 |
| MuSiQue | 1.396 | 1.787 | 3.320 |

**Precision and Optimization Details** All models were evaluated using single-precision floating-point arithmetic (FP32). For CE, `ListCE`, and `SetCE`, the maximum input sequence length was capped at 512 tokens. Before timing measurements, each benchmark session included 10 warmup iterations to stabilize GPU kernels and initialize runtime libraries, including FAISS.

**Retrieval Speed Analysis** Table 5 reports inference latency (seconds per query) for different retrieval and reranking models on 2WikiMultiHopQA, measured as a function of the candidate size $K$. Dense retrieval and query embedding costs are excluded from all measurements, allowing us to isolate the additional overhead introduced by each retrieval or reranking component.

The bi-encoder exhibits nearly constant latency across different values of $K$, as it performs no additional computation beyond candidate retrieval. In contrast, the cross-encoder (CE) shows a monotonic increase in inference latency as $K$ grows, reflecting its linear dependence on the number of candidate passages.

`ParaSet` introduces a modest additional overhead compared to the bi-encoder, but scales more favorably than CE due to its lightweight single-layer set scoring architecture. ListCE and SetCE incur substantially higher inference costs, as they require multiple forward passes over long input sequences, making them particularly sensitive to increases in $K$.

Hybrid pipelines combining `ParaSet` with ListCE or SetCE achieve a favorable balance between efficiency and effectiveness. By restricting reranking to a fixed number of top-ranked passage sets produced by `ParaSet`, the inference cost of the reranking stage remains bounded and grows only mildly with $K$. This allows the hybrid approach to retain much of the effectiveness of cross-encoder-based reranking while significantly reducing its computational overhead.

## E. Effect of Adjacent-Ranking Weight on Set Size Bias

This section provides a detailed analysis of how the weight of the fine-grained adjacent-ranking term ($\lambda$ in Eq. 1) influences the *set size bias* of `ParaSet`.

Recall that the training objective consists of a coarse ranking constraint that enforces the gold set to be ranked above all candidates, together with an adjacent-ranking term:

$$\mathcal{L}_{\text{hard}} = \mathcal{L}_{\text{pos-all}} + \lambda \, \mathcal{L}_{\text{adj}}.$$

The adjacent-ranking loss $\mathcal{L}_{\text{adj}}$ provides finer-grained supervision by contrasting candidate sets that differ only subtly in quality under a predefined set-quality ordering (recall-first and precision-second in our experiments). Increasing $\lambda$ therefore strengthens the pressure to separate nearby candidate sets generated through structured perturbations such as **addition**, **elimination**, and **interchange**, as well as negatives induced implicitly by in-batch sampling.

For each dataset, we train `ParaSet` with different values of $\lambda$ while keeping all other hyperparameters fixed. At inference time, we retrieve the top-20 documents using a bi-encoder and apply `ParaSet` to rank candidate passage sets constructed from this fixed pool. To quantify how the strength of adjacent-ranking supervision affects retrieval behavior, we measure the *average size of the top-ranked set* predicted by `ParaSet`. Here, the top-ranked set refers to the single highest-scoring candidate set under $S(q, \cdot)$ among all sets considered during inference. This metric captures whether training encourages the model to favor more *parsimonious* sets with fewer documents, or more *inclusive* sets that incorporate additional, potentially redundant evidence. Table 6 reports the average size of the top-ranked predicted set across three datasets. The effect of $\lambda$ is highly dataset-dependent. On **2WikiMultihopQA**, increasing $\lambda$ leads to a consistent decrease in the predicted set size, from 4.08 to 3.35. Given that the average number of gold supporting documents in this dataset is approximately 2.4, this trend suggests that stronger adjacent-ranking supervision helps prune non-essential documents while preserving coverage of relevant evidence.

In contrast, **HotpotQA** exhibits the opposite behavior: as $\lambda$ increases, the predicted set size grows sharply and saturates near the candidate limit. This suggests that, under HotpotQA's evidence distribution, overweighting adjacent-ranking constraints can bias the model toward overly inclusive sets. Finally, **MuSiQue** also shows an increasing trend in set size with larger $\lambda$, although the effect is more moderate.

Overall, these results highlight $\lambda$ as a critical hyperparameter that governs the trade-off between coverage and parsimony in set-level retrieval. Importantly, its optimal value is not universal but depends on dataset-specific properties such as the typical number of supporting documents and the prevalence of plausible yet irrelevant distractors.

