# OpenReview forum: "ParaSet: Retrieving Set of Passages in an Efficient Manner"
_ICML.cc/2026/Conference — Submitted to ICML 2026_

### Official Review · Reviewer_DZ6N · 2026-03-05

**Soundness:** 3
**Presentation:** 3
**Significance:** 3
**Originality:** 2
**Overall Recommendation:** 3
**Confidence:** 4

**Summary:**

The paper studies document retrieval, posing it as a set retrieval problem rather than ranking of individual documents. They focus on multi-hop QA setting, where retrieving a set of evidence passages is required. From what I understand, what they are proposing is a “reranker”, not a retriever. They use base retriever to get the top K set, and then introduce a single-layer attention-based model that operates on query and passage embeddings to decide identify the final candidate set. Compared to prior approaches which encoded query + documents with cross encoder, they use more efficient, compact representation, improving efficiency at the cost of accuracy.
The paper is clearly written and has some merits. The research question studied is valuable and understudied in the field, and evaluation is sound with various metrics (end-to-end QA performance, efficiency, etc). However, the experimental results are not very promising (baseline approaches mostly outperforms in task performance), and the evaluation scope is somewhat limited with limited/weak baselines.

**Compliance With Llm Reviewing Policy:**

Affirmed.

**Final Justification:**

I appreciate the author's response. I think the paper can be strengthened with more in-depth analysis on scaling behaviors.

**Key Questions For Authors:**

* I’m confused about the computational requirements for this methods in comparison to other baselines implemented. It might be good to have a table summarizing the inference costs (in terms of complexity). At the end of Page 4, it compares to other methods saying it’s cheaper than other beam-search-based retrieval methods. If I understand correctly, the proposed model also does beam-search based retrieval, just it is cheaper because it does not do cross-encoding (using pre-computed vector embedding per document).
* Can we consider long-context LLM for this? https://arxiv.org/abs/2406.13121 As the method works with subset of documents, I think this would work pretty well.
* How would it compare to approaches that handles set issues it at retrieval, not reranking phase? i.e., https://arxiv.org/abs/2511.02770
* Could this method scale to larger set of documents to be reranked? Experiments only shows up to K=30?

**Limitations:**

yes

**Strengths And Weaknesses:**

Strength:
* The paper is overall well-written and clear.
* The research problem it studies is valuable yet understudied in reviewers opinion.
* The evaluation considers variable aspects.

Weakness:
* The idea of using document embedding for efficient encoding (at the cost of expressivity) has been explored extensively. https://arxiv.org/pdf/2405.13792 While using it for retrieval reranking is new (as far as reviewer is aware of), I do not think the idea is very original / significant here.
* Weak baseline. The experiment relies on a single bi-encoder retriever model (Contriever) from 2022.   There are many updated retrievers, and this architectural choice makes the work weaker. How would it work when applied to newer models (e.g., InfRetriever, Qwen3-embedding models, or other strong ones on MTEB benchmark)? It’s much easier to show gains with weaker baseline models. Many of these models have the same API as contriver, so should be relatively easy to evaluate on. I also think it is important to run on 2+ base retrieval architecture to show that the method can be applied to various base retrievers.

---

> ### Author Rebuttal · Authors · 2026-03-31
>
> We thank the reviewer for the insightful and constructive feedback.
>
> ---
>
> **W1. Originality of embedding-based approaches (e.g., xRAG).**
>
> We agree that embedding-based representations are well-studied and do not claim novelty in this aspect.
>
> Our contribution addresses a different problem. Prior work such as xRAG focuses on **compressing large textual context into a single representation for decoding**, whereas our method focuses on **modeling and searching over combinations of documents**.
>
> This distinction is key:
> - xRAG: compress *single-source context* → efficient decoding
> - Ours: model *combinatorial document sets* → efficient set selection
>
> More importantly, our setting involves an **exponentially large set space**. Evaluating such combinations with LLM-based encoding would require repeated full-text processing, which is computationally infeasible. Our method enables scalable exploration of this space via efficient set-level scoring.
>
> Thus, the novelty lies in enabling **efficient set-level compatibility modeling and combinatorial search**, rather than in embedding usage itself.
>
> ---
>
> **W2. Evaluation on multiple base retrievers.**
>
> We agree and conducted additional experiments using a stronger retriever (**Qwen3-embedding-0.6B**).
>
> **Results (Qwen3-embedding-0.6B)**
>
> (HotpotQA)
>
> | Method | EM | F1 |
> |--------|----|----|
> | Bi | 0.3699 | 0.4730 |
> | Bi+CE | 0.3844 | 0.4935 |
> | Bi+SetCE | **0.3958** | **0.5079** |
> | Bi+ParaSet+SetCE | 0.3692 | 0.4801 |
>
> (2Wiki)
>
> | Method | EM | F1 |
> |--------|----|----|
> | Bi | 0.2851 | 0.3119 |
> | Bi+CE | 0.3100 | 0.3347 |
> | Bi+SetCE | 0.3075 | 0.3349 |
> | Bi+ParaSet+SetCE | **0.3472** | **0.3785** |
>
> (MuSiQue)
>
> | Method | EM | F1 |
> |--------|----|----|
> | Bi | 0.1260 | 0.2023 |
> | Bi+CE | 0.1335 | 0.2105 |
> | Bi+SetCE | **0.1513** | **0.2381** |
> | Bi+ParaSet+SetCE | 0.0894 | 0.1598 |
>
>
> These results show consistent trends across retrievers, indicating that our method is **retriever-agnostic**.
>
> ---
>
> **Q1. Computational requirements vs. cross-encoder baselines.**
>
> We agree this can be clarified.
>
> While both approaches use beam search, the key difference is **cost per set evaluation**:
> - Cross-encoder: full forward pass over concatenated text
> - Ours: lightweight scoring over pre-computed embeddings
>
> This enables evaluating many more candidate sets efficiently.
>
> Specifically, we adopt a two-stage pipeline (e.g., ParaSet+SetCE):
> 1. **ParaSet** performs fast set exploration
> 2. A stronger reranker (e.g., SetCE) is applied to a small subset
>
> Empirically (Table 5), ParaSet is **8×–30× faster**, and ParaSet+SetCE remains significantly faster than SetCE alone while achieving comparable performance.
>
> ---
>
> **Q2. Compatibility with long-context LLMs.**
>
> We agree that our method is complementary to long-context LLMs.
>
> Even with large context windows, document selection remains necessary, as real-world corpora (e.g., HotpotQA) far exceed context capacity. Moreover, figure 5 of this prior work shows that simply increasing context size can degrade retrieval quality.
>
> Our method addresses a complementary problem: selecting a **compact and globally coherent set of documents**. It can serve as a preprocessing module for long-context LLMs. We also expect that combining larger candidate pools (higher recall) with our set-level selection is a promising direction.
>
> ---
>
> **Q3. Comparison with retrieval-stage set/diversity methods.**
>
> We view retrieval-stage methods (e.g., multi-query retrieval) as **complementary**:
> - Retrieval-stage: improves **coverage/diversity**
> - Our method: ensures **global compatibility of selected sets**
>
> Diverse retrieval does not guarantee coherent evidence sets. Our method explicitly models interactions within the set. Combining both approaches is a natural direction, as our method benefits from stronger candidate pools.
>
> ---
>
> **Q4. Scalability to larger candidate sets (K > 30).**
>
> We agree this is important and conducted additional experiments on 2WikiMultiHopQA with K up to 70.
>
> | Model | K=10 | K=30 | K=50 | K=70 |
> |--------|------|------|------|------|
> | Bi | 0.2634 | 0.1182 | 0.0780 | 0.0583 |
> | SetCE | 0.5347 | 0.5554 | 0.5716 | 0.5501 |
> | ParaSet | 0.5131 | 0.4996 | 0.4847 | 0.4680 |
> | ParaSet+SetCE | 0.5566 | 0.5319 | 0.5479 | 0.5330 |
>
> Our method maintains **stable performance** as K increases, without significant degradation. ParaSet+SetCE remains competitive across all K.
>
> From an efficiency perspective, the latency increase of ParaSet+SetCE is modest (as referred in **Q1**), as ParaSet introduces only a lightweight overhead before reranking. This demonstrates that our method scales effectively to larger candidate sets.
>
> We will include these results in the final version.

---

> > ### Author Rebuttal · Reviewer_DZ6N · 2026-04-03
> >
> > Thanks for the response. I appreciate the new experimental results and thoughtful response.
> >
> > The new experimental results are good to see, but I'm not sure I agree with the interpretations of the authors... For example, of the Qwen embedding model, on MusiQue dataset, doing ParaSet introduces significant performance drop with this retriever Bi+SetCE 0.1513 -> Bi+ParaSet+SetCE	0.0894, while the drop was a lot less severe with the other retriever (0.0898 -> 0.0730)? Could you also present the Bi + ParaSet results?
> >
> > The scaling is also not as positive, as the gap between ParaSet and SetCE increases as K increases...
> >
> > On a slightly separate note, I think paper should be framed as reranker, not as a retriever, as it mainly deals with <100 documents, not the entire corpus.

---

> > > ### Author Response · Authors · 2026-04-08
> > >
> > > We thank the reviewer for the insightful feedback and follow-up questions.
> > >
> > > ---
> > >
> > > ### (1) On scaling behavior
> > >
> > > We thank the reviewer for this observation and agree that the gap between ParaSet and SetCE increases as K grows.
> > > We would like to clarify that ParaSet is designed to function similarly to a **bi-encoder at the set level**, and is not intended to achieve the highest accuracy on top-1 prediction. Instead, its role is to serve as a **fast set-level reranker** over the combinatorial space of $2^m$ candidate sets (given $m$ initial documents), identifying a small number of promising candidates.
> > > In this sense, ParaSet is an extremely lightweight model (less than 30MB of RAM) that enables efficient exploration of a very large search space. Its primary objective is to maintain **high recall within a small top-k candidate set**, rather than to directly select the optimal set.
> > > From this perspective, the observed drop in top-1 performance as search space grows ($2^k$ growth) as $k$ grows  is expected. As the number of possible sets increases exponentially, it becomes increasingly difficult for a lightweight model to identify the single best set.
> > > To better understand this behavior, we analyze Max-F1@$k_p$ while varying two parameters. Here, Max-F1@$k_p$ is defined as the maximum F1 score among the top-$k_p$ retrieved documents for each query:
> > > - $k_b$: number of initial document candidates retrieved by the bi-encoder
> > > - $k_p$: number of top candidate sets selected by ParaSet and passed to the reranker (e.g., SetCE)
> > >
> > >
> > > | $k_b$ \ $k_p$ | 1 | 3 | 5 | 10 | 20 | 30 | 40 | 50 |
> > > |---|---|--|--|--|--|--|--|--|
> > > | 5 | 0.228 | 0.248 | 0.264 | 0.272 | 0.274 | 0.274 | 0.274 | 0.274 |
> > > | 10 | 0.168 | 0.205 | 0.214 | 0.226 | 0.228 | 0.231 | 0.233 | 0.234 |
> > > | 20 | 0.159 | 0.203 | 0.214 | 0.226 | 0.238 | 0.239 | 0.239 | 0.242 |
> > > | 30 | 0.142 | 0.178 | 0.203 | 0.222 | 0.229 | 0.231 | 0.239 | 0.241 |
> > >
> > > Given fixed $k_p$, F1 decreases as $k_b$ increases due to the enlarged search space. However, **Max-F1 consistently improves as $k_p$ increases**, showing that ParaSet improves the **candidate pool quality**.
> > > In our ParaSet+SetCE experiments(Table 2), we fixed $k_p=10$. However, as the bi-encoder retrieves more documents, increasing $k_p$ becomes necessary to fully exploit the expanded pool. As shown in our additional experiments, increasing $k_p$ improves QA performance, and we expect similar gains in retrieval performance.
> > >
> > > ---
> > >
> > > ### (2) Retriever vs. reranker
> > >
> > > We agree the distinction can be nuanced.
> > >
> > > While our method operates on <100 documents, the **effective search space is combinatorial**. Given $m$ documents, there are $2^m$ possible sets. Even restricting to size $k$, the space remains large (e.g., $k=9 \rightarrow \sim 3.1 \times 10^9$).
> > >
> > > Since candidate sets are not explicitly enumerated, our method performs **search over a combinatorial set space**, rather than reranking a predefined list. For this reason, we used the term “retriever.”
> > >
> > > ---
> > >
> > > ### (3) Performance of Qwen embedding
> > > To address this point, we provide (1) an interpretation of the behavior on the MuSiQue dataset, and (2) additional experiments that further analyze this phenomenon.
> > >
> > > | (1) On the peculiarity of MUSIQUE
> > >
> > > Unlike other datasets, we noticed that the MuSiQue dataset has a very weak connection between documents, the connecting document was decided through the existence of an overlapping single entity.
> > > As a result, the semantic alignment between a document set and the query is inherently weak (e.g., documents on vastly different topics may still be selected due to sharing a single entity).
> > > Consequently, it becomes challenging to rank the gold document set highly when considering combinations.
> > > Therefore, as mentioned earlier, including the gold combination typically requires using a larger $k_p$. As shown in the table below, when we increase $k_p$, the resulting performance becomes competitive with SetCE beam search.
> > >
> > >
> > >
> > > | (2) Another result for ParaSet+SetCE on MuSiQue dataset with increased ParaSet’s Top-S candidate set number.
> > >
> > > To further investigate this behavior, we conducted additional experiments by increasing the number of candidate sets passed from ParaSet to SetCE.
> > > In our default setting, $k_p = 10$. We increase this value to allow SetCE to rerank a larger set of candidates to $k_p = 50$.
> > > As in the Table in (1) of Reviewer iV54’s post-rebuttal response, this modification does not introduce significant computational overhead, as ParaSet is lightweight and the additional cost remains moderate.
> > >
> > > | Method  | EM   | F1  |
> > > |--|--|--|
> > > | CE    | 0.1335 | 0.2105 |
> > > | SetCE  | 0.1513 | 0.2381 |
> > > | ParaSet+SetCE ($k_p=10$)    | 0.0894 | 0.1598 |
> > > | ParaSet+SetCE ($k_p=50$)    | 0.1424 | 0.2406 |
> > >
> > >
> > > We observe that increasing $k_p$ improves QA performance. Notably, this allows ParaSet+SetCE to achieve performance that is competitive with SetCE alone, while still requiring significantly less computation than running SetCE over a large candidate space directly.

---

### Official Review · Reviewer_DV3C · 2026-03-13

**Soundness:** 3
**Presentation:** 3
**Significance:** 3
**Originality:** 3
**Overall Recommendation:** 4
**Confidence:** 4

**Summary:**

This paper revisits multi-hop retrieval as a set-level scoring problem rather than a sequence of local next-passage decisions. The proposed approach combines a margin-based set-level compatibility objective with a lightweight single-layer attention scorer, ParaSet, that can efficiently score many candidate passage sets during beam search. Experiments on HotpotQA, 2WikiMultihopQA, and MuSiQue suggest that set-level supervision improves robustness over a sequential list-level retriever and yields a promising quality-efficiency trade-off.

**Compliance With Llm Reviewing Policy:**

Affirmed.

**Final Justification:**

The rebuttal addresses my concerns more clearly, and I appreciate the additional clarifications and follow-up experiments, especially regarding the stronger retriever analysis and the budget-controlled QA comparison. Therefore, I will keep my score unchanged.

**Key Questions For Authors:**

1. Can the authors provide a budget-matched version of Table 2 where all methods rerank the same number of retrieved candidates?

2. How should readers reconcile the fixed maximum set size of 9 in Appendix D with the average predicted set sizes of 19.80 and 19.88 reported in Table 6?

3. Can the authors test whether a stronger or hybrid first-stage retriever mitigates the performance degradation observed on MuSiQue as K increases?

4. Can the authors provide the prompt and decoding details for the `gpt-4o-mini` end-to-end QA evaluation, and indicate whether the numbers are from a single run or averaged over multiple runs?

**Limitations:**

The paper briefly mentions that the method depends on the quality of the initial candidate pool and that some behaviors are dataset-dependent, but the limitation discussion is still fairly brief. It would be helpful to discuss more explicitly that the method cannot recover supporting passages that are missed by the first-stage retriever, and that this issue may become more visible on harder datasets such as MuSiQue.

**Strengths And Weaknesses:**

Strengths:
1. The paper addresses a meaningful weakness of sequential multi-hop retrieval, namely that local supervision does not directly optimize the quality of the full evidence set.
2. The `ListCE` versus `SetCE` comparison is informative because it isolates the effect of the proposed set-level objective while keeping architecture and inference fixed.
3. The paper considers both quality and efficiency, which is appropriate for a practical retrieval method.

Weaknesses:
1. The main end-to-end QA comparison is not fully matched across methods. In Table 2, CE/ListCE/SetCE operate on 20 retrieved candidates, while ParaSet-based methods are allowed to search over 50. This does not negate the empirical gains, but it makes the comparison harder to interpret cleanly.
2. There is an appendix inconsistency that should be clarified. Appendix D states that the maximum retrieved set size is fixed to 9 for all beam-search-based methods, but Table 6 reports average predicted set sizes of 19.80 and 19.88 on HotpotQA.
3. The method appears somewhat sensitive to the quality of the initial bi-encoder candidate pool. The MuSiQue analysis suggests that adding more candidates can hurt when the first-stage retriever is weak, but the paper does not test whether a stronger or hybrid initial retriever would mitigate this issue.
4. The downstream QA evaluation with `gpt-4o-mini` is somewhat under-specified. The few-shot prompt, decoding settings, and run-to-run variability are not described in enough detail.

Minors:
There are also a few minor writing issues that should be corrected, such as "an lightweight", "a more decline", and "Experiment is conducted".

---

> ### Author Rebuttal · Authors · 2026-03-31
>
> We thank the reviewer for the careful reading and constructive feedback.
>
> ---
>
> **W1. The end-to-end QA comparison is not fully matched because ParaSet-based methods use a larger candidate pool.**
>
> We thank the reviewer for this insightful comment.
>
> We would like to clarify that our experimental budget was defined based on two criteria:
> (1) comparable inference cost (e.g., runtime), and
> (2) using hyperparameters that allow each method to achieve its **best performance**.
>
> In this sense, our goal was not to enforce identical candidate sizes (i.e., the same K), but rather to provide a fair comparison where each method operates under its most effective configuration. In particular, ParaSet is a lightweight model (≈30MB) with significantly lower per-set cost, which allows it to explore a larger candidate space within a similar computational budget.
>
>
> We agree that fixing K across all methods could provide an additional perspective. Due to time constraints, we were not able to include this analysis in the rebuttal. However, we will include a budget-matched comparison (e.g., fixed K setting) in the camera-ready version to provide a clearer and more controlled comparison.
>
>
> ---
>
> **W2. Appendix D states a maximum set size of 9, but Table 6 reports average predicted set sizes of 19.80 / 19.88.**
>
> Thank you for catching this inconsistency.
>
> The values in Table 6 correspond to the size of the **top-1 predicted set produced by ParaSet** in an auxiliary analysis where we intentionally removed the maximum set-size constraint, in order to better reflect the model’s unconstrained behavior.
>
> We agree that this setting was not clearly described. We will revise the final version to explicitly state this setup and remove the ambiguity.
>
> ---
>
> **W3. Can a stronger or hybrid first-stage retriever mitigate the MuSiQue degradation as \(K\) increases?**
>
> We agree this is an important question. As we understand it, this concern mainly refers to the retrieval-performance trend in Figure 2, rather than the end-to-end QA results.
>
> Unfortunately, we were not able to complete this specific retrieval-side experiment during the rebuttal period. We plan to run this analysis for the final version.
>
> If the reviewer’s question also concerns **end-to-end QA**, we would like to point to our additional experiment in response to Reviewer DZ6N, where we replaced Contriever with **Qwen3-embedding-0.6B** as the first-stage retriever. Under this stronger retriever, MuSiQue performance improved over the Contriever-based setup, suggesting that a better candidate pool may indeed alleviate the degradation. We will verify this more directly in retrieval-space analysis in the final version.
>
> ---
>
> **W4. Prompt and decoding details for GPT-4o-mini QA evaluation.**
>
> We are happy to clarify this setup.
>
> All reported end-to-end QA results were obtained with a single run using the following configuration:
>
> - **Model:** GPT-4o-mini
> - **Temperature:** 0.2
> - **Max output tokens:** 1024
> - **Few-shot setting:** 5 in-context examples per dataset
>
>
> **Prompt:**
> The user message contains:
> 1. **5 few-shot examples** from the same dataset; followed by
> 2. the actual evaluation instance, consisting of the retrieved snippets and the question.
>
> A simplified template is:
>
> ```text
> [Few-shot examples]
>
> Title: [Example Title]
> [Example Content]
> Q: [Example Question]
> A: [Example reasoning... So the answer is: Example Answer.]
>
> ...
>
> [Evaluation instance]
>
> Title: [Retrieved Document Title 1]
> [Retrieved Document Snippet 1]
> Title: [Retrieved Document Title 2]
> [Retrieved Document Snippet 2]
> ...

---

> > ### Author Rebuttal · Reviewer_DV3C · 2026-04-02
> >
> > Overall, the rebuttal basically addresses my concerns. I appreciate the clarifications regarding the Table 6 inconsistency and the GPT-4o-mini QA evaluation details. However, it is unfortunate that the additional experiments for W1 and W3 could not be included during the rebuttal period due to time constraints. Therefore, while my assessment is more positive after reading the rebuttal, I will keep my score unchanged.

---

> > > ### Author Response · Authors · 2026-04-08
> > >
> > > We thank the reviewer for the positive feedback and constructive suggestions.
> > >
> > > ---
> > >
> > > ### (1) On W3: Stronger retriever and MuSiQue degradation
> > >
> > > Regarding W3, while we were not able to include the full experiment during the initial rebuttal period, we have now conducted an additional analysis using a stronger retriever (**Qwen3-embedding-0.6B**), following the reviewer’s suggestion.
> > >
> > > The table below reports retrieval F1 scores for $k=5$ to $k=30$, along with the corresponding performance change ratios.
> > >
> > > | Retriever       | Model               | K5     | K10    | K15    | K20    | K25    | K30    | $\Delta$F1(%)  (k=30 v.s. k= 5) |
> > > |----|----|--|--|--|--|--|--|--|
> > > | Contriever | ParaSet        | 0.2281 | 0.1680 | 0.1580| 0.1590 | 0.1430| 0.1420| -0.377   |
> > > | Contriever | ParaSet + SetCE     | 0.2390 | 0.1904 | 0.1843 | 0.1759 | 0.1704 | 0.1628 | -0.319            |
> > > | Contriever | SetCE               | 0.1375 | 0.1731 | 0.1532 | 0.1597 | 0.1719 | 0.1735 | 0.262           |
> > > | Qwen3      | ParaSet             | 0.3150 | 0.2948 | 0.2685 | 0.2659 | 0.2668 | 0.2613 | -0.170            |
> > > | Qwen3      | ParaSet + SetCE     | 0.3840 | 0.3291 | 0.3188 | 0.3149 | 0.2978 | 0.2926 | -0.238            |
> > > | Qwen3      | SetCE               | 0.2787 | 0.3632 | 0.3503 | 0.3458 | 0.3306 | 0.3537 | 0.269           |
> > >
> > > We observe that, with a stronger retriever, the performance drop is **consistently reduced** across ParaSet, ParaSet+SetCE, and SetCE. This suggests that improving the quality of the initial top-$k$ candidates can partially mitigate the robustness issue observed in Figure 2.
> > > However, we also note that as $k$ increases, a performance drop still remains, similar to what is observed with Contriever. This indicates that while a stronger retriever helps, it does not fully resolve the issue, and the underlying challenge remains.
> > >
> > > Importantly, we would like to emphasize that ParaSet functions as a set-level retriever. As shown in Reviewer DZ6N’s post-rebuttal comment (1), even if the top-1 set performance is not the highest, ParaSet plays a **crucial role by retrieving top-K candidate sets** and passing them to SetCE.
> > > The comparison between k=5 and k=30 in the table above reflects the performance of the top-1 retrieved set for each model. As the number of documents retrieved by the bi-encoder (denoted as k) increases, the combinatorial search space over sets grows rapidly. In this regime, selecting a larger number of candidate sets and leveraging ParaSet+SetCE becomes particularly beneficial. As demonstrated in Reviewer DZ6N’s post-rebuttal comment (3) on Qwen embeddings, this approach achieves performance comparable to using SetCE alone, while requiring significantly less computational cost.
> > >
> > > ---
> > >
> > > ### (2) On W1: Budget-matched QA comparison
> > >
> > > We further conduct a **budget-controlled comparison** by varying the number of retrieved documents:
> > >
> > > | Model   | EM  | F1  |
> > > |--|--|--|
> > > | CE (k=20)   | 0.0853 | 0.1588 |
> > > | CE (k=50, new)   | 0.0874 | 0.1626 |
> > > | ListCE (k=20)   | 0.0807 | 0.1543 |
> > > | ListCE (k=50, new)   | 0.0825 | 0.1531 |
> > > | SetCE (k=20) | 0.0898 | 0.1625 |
> > > | SetCE (k=50, new) | 0.0848 | 0.1627 |
> > > | ParaSet (k=50)  | 0.0621 | 0.1294 |
> > > | ParaSet+ListCE (k=50) | 0.0710 | 0.1375 |
> > > | ParaSet+SetCE (k=50)  | 0.0730 | 0.1406 |
> > >
> > > We observe that increasing the candidate pool does not uniformly improve performance, and the effect is **method-dependent**, making direct one-to-one comparisons incomplete.
> > >
> > > This suggests that the effect of a larger candidate pool is method-dependent, and that a simple eye-to-eye comparison between CE and ParaSet+SetCE may not fully capture the role of each method.
> > >
> > > To better understand this, we analyze the QA prediction overlap with CE for each method (M):
> > >
> > > | Method (M) | M \ CE (%) | CE \ M (%) | M ∩ CE (%) | M ∪ CE → H (%) |
> > > |--|--|--|--|--|
> > > | ParaSet  | 21   | 30  | 49 | 74  |
> > > | ParaSet+ListCE | 22 | 27  | 51 | 78.2  |
> > > | ParaSet+SetCE | 22  | 28  | 50  | 79.4  |
> > > | ListCE  | 17 | 23  | 60  | 68.5   |
> > > | SetCE  | 16  | 22  | 62  | 70.3 |
> > >
> > > Here, **M \ CE** denotes cases where the model is correct but CE is not,
> > > **CE \ M** denotes the opposite, and **M ∩ CE** denotes cases where both are correct.
> > > We further define **M ∪ CE → H** as the proportion of questions that are correctly solved by the **hybrid model (H)** (i.e., use the union of top-1 set of M and top-5 documents of CE, as an “Additional Experiment” in reviewer ntpG’s rebuttal) among those solved by either M or CE.
> > >
> > > ParaSet-based methods show:
> > > - **lower overlap with CE (low M ∩ CE)**
> > > - **higher M-CE**, i.e., solving more cases where CE fails
> > >
> > > This indicates that ParaSet captures **complementary subsets of questions**, unlike ListCE/SetCE which behave more similarly to CE.
> > >
> > > This is further supported by higher **M ∪ CE → H**, showing stronger recovery of complementary cases.
> > >
> > > ---
> > >
> > > ### Summary
> > >
> > > Overall, ParaSet is best understood as a **complementary set-level retrieval mechanism** that enhances performance when combined with CE-style methods.

---

### Official Review · Reviewer_iV54 · 2026-03-13

**Soundness:** 2
**Presentation:** 3
**Significance:** 3
**Originality:** 2
**Overall Recommendation:** 3
**Confidence:** 3

**Summary:**

The paper focuses on the need to select a set of passages for multi-hop question answering. Instead of selecting independent passages as in single-turn retrieval or forming a sequential dependency of multi-round retrieval, the paper focuses on scoring the relevance between a query and a set of passages. The paper then proposes ParaSet, which includes set-level compatibility learning and a lightweight neural architecture. During inference, to avoid the computational cost of set combinations, the paper proposes using beam search combined with a set scoring function. The method shows a good balance between retrieval performance and efficiency.

**Compliance With Llm Reviewing Policy:**

Affirmed.

**Final Justification:**

The authors partially resolved my concerns. However, the answer to W1 actually shows that Bi+CE is better solution than ParaSet. The improvement on sequential is modest, whereas the drop on parallel is significant (74.00>62.35 on HotpotQA) and (75.00>53.62 on 2Wiki). A robust set evaluation should maintain the performance on parallel, while enhancing the performance on sequential data. In practice, we can't know for sure which type of question is parallel, which one is sequential.

In addition, with Beamsearch not being the contribution of this paper, the method is not particularly exciting. The set scoring, though being more efficient, is not much effective. Paraset still needs to rely on the expensive SetCE for obtaining best performance.

For such reasons, I will keep my score unchanged.

**Key Questions For Authors:**

Please check the weaknesses

**Limitations:**

No.

**Strengths And Weaknesses:**

*Strengths*:

- The motivation is grounded in empirical support, analyzing the limitations of sequential retrieval.
- The paper is well written in general.
- The experiments are comprehensive.

*Weaknesses*:
- As shown in Table 2, standalone ParaSet yields consistently worse end-to-end QA performance than even the basic passage-level Bi+CE baseline across all three benchmarks. Meanwhile, the ParaSet+CE/SetCE hybrid pipeline, despite delivering marginal performance improvements, incurs drastically higher inference latency than the standalone CE baseline (as evidenced in Figure 2). This raises the question of whether such complicated optimization is necessary if a simple Bi+CE works. A deeper analysis with case studies is needed to explain why Bi+CE shows lower retrieval performance (Figure 2) yet obtains better downstream QA results.
- The idea of Beam Search has been exploited in the work of Zhang et al. (2024), which lowers the novelty of this paper. In addition, in Section 3.2, the authors should explicitly state how their Beam Search is inspired by (and different from) that of Zhang et al.
- The predefined ordering function mentioned in Section 3.1 requires more clarification. How is this function defined? Furthermore, how does one compare sets that are perturbed from the gold set?
- In the related work section, the paper explicitly identifies Lee et al. (2025) and Chen et al. (2025) (Sections 2.3 and 4) as the most central prior works on set-level retrieval within the same research area. Both adopt the same research paradigm as this paper—focusing on set-level modeling for multi-hop retrieval—and represent the current state-of-the-art methods in this line of work. However, the experimental section does not conduct head-to-head comparisons in terms of performance and inference efficiency with these two methods.

---

> ### Author Rebuttal · Authors · 2026-03-31
>
> We thank the reviewer for the detailed and constructive feedback.
>
> ---
>
> **W1. Is such a complex method necessary if Bi+CE already works well?**
>
> This question highlights an important distinction.
> Our method is particularly beneficial when solving a query requires **modeling interactions among documents**, rather than treating them independently.
>
> To analyze this, we categorize queries into:
> - **Parallel**: documents independently contribute to the answer
> - **Sequential**: documents must be composed in a chain
>
> Both HotpotQA and 2Wiki contain such variations within 2-hop questions, enabling a controlled comparison.
>
> **QA F1 across reasoning structures**
>
> | Dataset | Category | CE | SetCE | ParaSet | ParaSet+SetCE |
> |--|--|--|--|--|--|
> | Hotpot | parallel | 74.00 | **75.13** | 62.35 | 71.11 |
> | Hotpot | sequential | 47.55 | 48.13 | 46.01 | **49.77** |
> | 2Wiki  | parallel | **75.00** | 69.74 | 53.62 | 55.47 |
> | 2Wiki  | sequential | 16.05 | 18.36 | 19.80 | **20.01** |
>
> **Key observations:**
> - In **parallel settings**, **Cross-Encoder (CE)** performs best.
>   This is expected, as CE is highly effective at modeling *query–document* relevance.
> - In contrast, in **sequential settings**, **set-based methods (SetCE, ParaSet+SetCE)** consistently outperform CE.
>   This indicates that modeling *query–set compatibility* is crucial when inter-document dependencies matter.
> - Notably, **ParaSet+SetCE achieves the best performance in sequential settings across both datasets**, suggesting that combining efficient set exploration (ParaSet) with expressive scoring (SetCE) is particularly effective.
>
> In fact, as discussed in our response to Reviewer ntpG (Additional Experiment), combining complementary methods leads to stronger performance.
>
> This explains when additional modeling is necessary beyond Bi+CE.
>
> ---
>
> **W2. Beam search is not novel (Zhang et al., 2024).**
>
> We thank the reviewer for the opportunity to clarify this point.
> We would like to emphasize that we do not claim beam search itself as a novel contribution. Rather, our contribution lies in enabling efficient set-level scoring, which makes beam search over the combinatorial set space practical.
>
> Specifically, ParaSet:
> - avoids expensive cross-encoding
> - enables lightweight exploration, followed by optional reranking
>
> As shown in Table 5, this design leads to substantial efficiency gains. Compared to ListCE/SetCE, the ParaSet+ListCE/SetCE pipeline achieves a 4×–9× speedup as the number of candidates K increases, while ParaSet alone achieves up to 8×–30× speedup.
>
> ---
>
> **W3. Clarification of the ordering function.**
>
> We rank candidate sets using **lexicographic ordering** based on recall and precision w.r.t. the gold set $G_q$:
>
> $\text{recall}(S) = \frac{|S \cap G_q|}{|G_q|}, \quad \text{precision}(S) = \frac{|S \cap G_q|}{|S|}$
>
> Sets are compared by recall first, then precision.
> This naturally orders perturbed sets (e.g., removing gold passages lowers recall; replacing them lowers both recall and precision), which we use to construct training pairs.
> We will clarify this with examples in the final version.
>
> ---
>
> **W4. Lack of comparison with prior set-level retrieval methods.**
>
> We thank the reviewer for this important suggestion. To address this concern, we conducted additional experiments to directly compare our method with SETR (Lee et al., 2025), which represents a representative set-level retrieval approach.
>
> |**Experimental setup clarification**
>
> We note that the original SETR paper fine-tunes a Llama-3.1-8B-Instruct model for set retrieval. In contrast, for a lightweight comparison, we use the GPT-4o-mini API (same as the question-answering model) **without any additional fine-tuning**, directly applying it for set selection.
>
> | **Performance comparison**
>
>  **HotpotQA**
> | Method | EM | F1 |
> |--|--|--|
> | Bi+CE | 0.390 | 0.506 |
> | Bi+SetCE | 0.391 | 0.506 |
> | Bi+ParaSet+SetCE | **0.404** | **0.522** |
> | SETR (bi+LLM) | 0.401 | 0.514 |
>
> **2WikiMultiHopQA**
> | Method | EM | F1 |
> |--|--|--|
> | Bi+CE | 0.284 | 0.309 |
> | Bi+SetCE | 0.241 | 0.267 |
> | Bi+ParaSet+SetCE | 0.287 | **0.332** |
> | SETR (bi+LLM) | **0.297** | 0.323 |
>
> **MuSiQue**
> | Method | EM | F1 |
> |--|--|--|
> | Bi+CE | 0.085 | 0.159 |
> | Bi+SetCE | 0.090 | **0.162** |
> | Bi+ParaSet+SetCE | 0.073 | 0.141 |
> | SETR (bi+LLM) | **0.091** | 0.143 |
>
> |**Efficiency comparison** (based on Llama-3.1-8B model)
>
> | Method | RAM Memory (GB) | Latency (sec/query) |
> |--|--|--|
> | SETR (LLM-based) | 15.32 | 2.10 |
> | SetCE | **2.12**  | 0.70 |
> | ParaSet+SetCE | 2.15 | **0.24** |
>
> |**Discussion**
> Across all benchmarks, our method achieves **competitive or superior performance** compared to SETR.
> Importantly, our approach is **substantially more efficient** in both memory and inference latency. Our method shows similar accuracy  compared to SetR while only using 15% of memory and 8x faster than SetR. This is because SETR (and similarly Chen et al., 2025) rely on large language models for retrieval.

---

> > ### Author Rebuttal · Reviewer_iV54 · 2026-04-03
> >
> > I would like to thank the authors for the response. The authors partially resolved my concerns. However, the answer to W1 actually shows that Bi+CE is better solution than ParaSet. The improvement on sequential is modest, whereas the drop on parallel is significant (74.00>62.35 on HotpotQA) and (75.00>53.62 on 2Wiki). A robust set evaluation should maintain the performance on parallel, while enhancing the performance on sequential data. In practice, we can't know for sure which type of question is parallel, which one is sequential.
> >
> > In addition, with Beamsearch not being the contribution of this paper, the method is not particularly exciting. The set scoring, though being more efficient, is not much effective. Paraset still needs to rely on the expensive SetCE for obtaining best performance.
> >
> > For such reasons, I will keep my score unchanged.

---

> > > ### Author Response · Authors · 2026-04-08
> > >
> > > We thank the reviewer for the constructive feedback.
> > >
> > >
> > > ---
> > >
> > > ### (1) Paraset still needs to rely on the expensive SetCE for obtaining best performance.
> > >
> > > We appreciate this observation and would like to provide further clarification.
> > >
> > > We first want to clarify that, as the number of candidate documents/sets increases, SetCE’s “single” forward pass is that of CE as shown in the following table, which reports the reranking time (excluding bi-encoder) for different number of candidates, where batching is performed to the maximum extent possible within the memory constraints of a single 24GB RTX 4090 GPU.
> > >
> > > | # Candidates (reranking) | 10     | 30     | 50     | 70     |
> > > |--------------------------|--------|--------|--------|--------|
> > > | CE (documents)           | 0.0219 | 0.0552 | 0.0937 | 0.1313 |
> > > | ListCE/SetCE (sets) | 0.0738 | 0.1014 | 0.1179 | 0.1365 |
> > >
> > > To clarify, this table does not report the runtime of SetCE with beam search. Rather, it measures the cost of individual forward passes of SetCE when evaluating candidate sets within the ParaSet+SetCE framework.
> > > Given this fact, we do not consider the requirement for reranking itself to be a drawback. In many retrieval systems, it is common to combine complementary components. For example, **bi-encoder + cross-encoder pipelines**, where the bi-encoder provides fast retrieval with high recall, and the cross-encoder refines the results with more accurate but slower scoring.
> > > We view our approach in a similar way. ParaSet serves as a **lightweight set-level explorer**, while SetCE provides more expressive scoring over a reduced candidate space. Their complementary design allows us to balance efficiency and accuracy similar to the use case of BE+CE.
> > >
> > >
> > > ---
> > >
> > > ### (2) On W1: CE outperforming ParaSet in parallel settings
> > >
> > > We agree that CE achieves stronger performance in parallel settings. However, ParaSet is not designed to replace CE as a final decision module, but to serve as a **lightweight set-level explorer** over a combinatorial space ($2^m$), prioritizing efficient exploration and high recall rather than top-1 accuracy.
> > >
> > > We also agree that query types (parallel vs. sequential) are not known in practice. This motivates a **complementary approach** rather than choosing one method over the other.
> > >
> > > Based on this insight, we conducted additional experiments combining CE with ParaSet-based methods (i.e., CE ∪ ParaSet+SetCE, as reported in our response of "Additional Experiment” to Reviewer ntpG). These results show that such a hybrid approach achieves the best performance with only a modest additional computational cost.
> > >  We therefore view the W1 analysis as evidence of **complementary strengths**, rather than a weakness of ParaSet.
> > >
> > >
> > > ---
> > >
> > > ### (3) On beam search contribution
> > >
> > > We appreciate the reviewer’s perspective and agree that performing beam search itself is not the  **main** contribution of this work.
> > > However, we would like to clarify that our contribution lies in several aspects beyond the beam search itself:
> > >
> > > - **Novel aspect of our beam search “usage”.**
> > > While beam search itself is a standard technique, our contribution lies in how it is utilized. Unlike prior work such as Zhang et al., which selects beam paths based on a threshold and retains only a subset of final trajectories, our approach preserves all intermediate sets encountered during beam search as candidates for downstream scoring.
> > > This design enables ParaSet to evaluate a significantly richer and more diverse set of candidate combinations, rather than being restricted to final beam outputs. In other words, beam search in our framework serves not merely as a pruning mechanism, but as a structured exploration strategy for constructing a candidate set pool. This aspect is detailed in Algorithm 1 (Appendix B), and will be further clarified in the camera-ready version.
> > >
> > > - **Set-level learning objective.**
> > > We demonstrate that set-level contrastive learning (SetCE) is more robust than list-wise training (ListCE), as shown in Figure 1. And therefore, SetCE often shows better end-to-end performance in QA when combined with ParaSet as well.
> > >
> > > - **Set-space modeling framework.**
> > > We introduce the notion of bi-encoder (BE) and cross-encoder (CE) analogues in the *set space*. To our knowledge, this is the first work that enables **lightweight set-level retrieval over a combinatorial space**, rather than relying on LLM-based generation or sequential list-wise modeling.
> > >
> > > Overall, we believe that the combination of **set-level modeling, efficient combinatorial search, and complementary integration with existing methods** constitutes the main contribution of this work.

---

### Official Review · Reviewer_ntpG · 2026-03-13

**Soundness:** 4
**Presentation:** 4
**Significance:** 3
**Originality:** 4
**Overall Recommendation:** 5
**Confidence:** 4

**Summary:**

The authors study retrieval tasks where the goal is to retrieve a set of multiple passages together. In the literature, this is typically approached with explicit multi-hop retrieval systems, which are trained to fetch independently relevant passages in each hop, and to depend in the generation of text queries or embeddings on previous hops, for fetching additional missing documents. To the contrary of this, the authors instead explore a (complementary in principle) question: can we design and train an efficient modeling approach to directly score and rank collective sets of documents?

The authors introduce a single-layer self-attention architecture that runs on top of standard bi-encoder representations of the set of passages. The authors supervise this architecture for scoring the compatibility between a query and a set of passages by constructing a candidate pool C of _document sets_ for each query. This pool includes the gold evidence set as well as sets produced by perturbations and in-batch negative sets. At search time, the authors employ a "retrieve singletons first" heuristic and then essentially re-rank possible sets.

The method proposed is not generally superior to explicit multi-hop search (at least not typically), but is substantially more effective than other single-step retrievers especially on 2WikiMultiHopQA, while still having good latency.

**Compliance With Llm Reviewing Policy:**

Affirmed.

**Final Justification:**

The authors clarifications reinforced my confidence in my strong evaluation of this interesting work. I can't justify raising the score or confidence further given the nature and substance of the work and its evaluations and my already strong stance, but I will maintain them.

**Key Questions For Authors:**

See above.

**Limitations:**

See above.

**Strengths And Weaknesses:**

This is a very well-motivated and written paper. The goal formulated (to learn set-level relevance scores) is understudied, and the architecture that the authors introduce is a simple and very interesting approach, which they train in a convincing manner. The results are strong in my perspective, even though I'm surprised the authors don't make the case more explicitly for composing this method with "agentic" multi-hop retrieval approaches. They seem complementary, as the authors also suggest. My main concern is that it's not obvious how the significance of this work should be understood. Is this a cool intellectual point in the tradeoff space that has been underexplored so far? I believe it is. But does it have much practical significance yet, or is it just a step towards richer methods in the future? In particular, with the current latency increases here, is the idea that this method competitive with multi-hop systems once we consider the tradeoff between latency and quality? Another aspect of the discussion that's missing is that many multi-hop queries are truly order-depend in a way that might be even render the "retrieve singletons first" heuristic structurally too weak to work without sequential hops. This is confounded right now by the fact that evaluation tasks mix such queries with genuinely hard bridge queries.

---

> ### Author Rebuttal · Authors · 2026-03-31
>
> We sincerely thank the reviewer for the positive assessment and thoughtful comments.
>
> |**Comment 1.** *Compositionality with agentic multi-hop retrieval seems complementary.*
>
> We agree. Agentic retrieval explores documents through sequential decisions, while our method directly scores **global query–set compatibility**. Because of this difference, the two approaches are naturally complementary: our method may be advantageous when explicit query decomposition is difficult, whereas agentic pipelines may be preferable when decomposition is effective.
>
> Our comparison against multi-step retrieval was intended to test whether set-level retrieval can serve as a competitive alternative to repeated sequential search. As shown in Section 4.4, our method is competitive with, and in some cases superior to, agentic retrieval pipelines while offering substantially lower latency. We also believe tighter integration is promising future work—for example, applying set-level reranking over documents collected by an agentic pipeline.
>
> |**Comment 2.** *Is this mainly an interesting intellectual point, or does it already have practical significance?*
>
> We believe it has practical significance as well. In our end-to-end QA results, **Bi+SetCE is 2–3× faster than agentic pipelines while achieving stronger QA performance**, indicating that set-level retrieval is not only conceptually interesting but also competitive in the latency–quality tradeoff.
>
> |**Comment 3.** *Some multi-hop queries are strongly order-dependent, so "retrieve singletons first" may be structurally weak.*
>
> We thank the reviewer for raising this important point.
> We agree that for certain multi-hop queries with strong order dependency, the *“retrieve singletons first”* paradigm may be structurally limited.
>
>
> However, as shown in **W1 of our response to Reviewer iV54**, even for chain-like (bridge-style) queries where documents are sequentially connected, **set-based models outperform cross-encoder baselines**.
> This suggests that while the limitation highlighted by the reviewer is valid, our approach remains more robust than existing methods under such conditions.
>
>
>
>
> **[Additional Experiments: Combining CE with Set-based Methods]**
>
>
> To further investigate this point and demonstrate complementarity across methods, we conduct an additional experiment:
>
>
> - **CE**: expand top-5 → top-10 documents
> - **SetCE / ParaSet+SetCE**: union of (top-1 set) + (CE top-5 documents)
>
>
> **Table 1: Results with CE augmentation (Contriever)**
>
> (Hotpot)
>
> | Model | EM | F1 | Original EM | Original F1 |
> |------|----|----|-------------|-------------|
> | bi+ce | 0.3977 | 0.5153 | 0.3903 | 0.5059 |
> | bi+SetCE | 0.3986 | 0.5167 | 0.3910 | 0.5058 |
> | bi+ParaSet | 0.3978 | 0.5126 | 0.3661 | 0.4788 |
> | bi+ParaSet+SetCE | **0.4095** | **0.5294** | 0.4041 | 0.5222 |
>
>
> (2Wiki)
>
> | Model | EM | F1 | Original EM | Original F1 |
> |------|----|----|-------------|-------------|
> | bi+ce | 0.3153 | 0.3526 | 0.2838 | 0.3090 |
> | bi+SetCE | 0.3198 | 0.3586 | 0.2407 | 0.2668 |
> | bi+ParaSet | 0.3390 | 0.3677 | 0.2631 | 0.2902 |
> | bi+ParaSet+SetCE | **0.3511** | **0.3914** | 0.2867 | 0.3315 |
>
> (MuSiQue)
>
> | Model | EM | F1 | Original EM | Original F1 |
> |------|----|----|-------------|-------------|
> | bi+ce | 0.0889 | 0.1475 | 0.0853 | 0.1588 |
> | bi+SetCE | 0.0891 | 0.1508 | 0.0898 | 0.1625 |
> | bi+ParaSet | 0.0784 | 0.1385 | 0.0621 | 0.1294 |
> | bi+ParaSet+SetCE | **0.0930** | **0.1567** | 0.0730 | 0.1406 |
>
>
> ---
>
>
> Following Reviewer DZ6N’s suggestion, we also evaluate a stronger retriever (**qwen3-embedding-0.6B**).
>
>
> **Table 2: Results with stronger retriever (Qwen3-embedding-0.6B)**
>
> (Hotpot)
>
> | Model | EM | F1 | Original EM | Original F1 |
> |------|----|----|-------------|-------------|
> | bi+ce | 0.4001 | 0.5164 | 0.3844 | 0.4935 |
> | bi+SetCE | 0.4045 | 0.5241 | 0.3958 | 0.5079 |
> | bi+ParaSet | 0.4078 | 0.5271 | 0.3094 | 0.4076 |
> | bi+ParaSet+SetCE | **0.4292** | **0.5531** | 0.3692 | 0.4801 |
>
> (2Wiki)
>
> | Model | EM | F1 | Original EM | Original F1 |
> |------|----|----|-------------|-------------|
> | bi+ce | 0.3429 | 0.3703 | 0.3100 | 0.3347 |
> | bi+SetCE | 0.3378 | 0.3650 | 0.3075 | 0.3349 |
> | bi+ParaSet | 0.3390 | 0.3677 | 0.1479 | 0.1719 |
> | bi+ParaSet+SetCE | **0.3527** | **0.3797** | 0.3472 | 0.3785 |
>
> (MuSiQue)
>
> | Model | EM | F1 | Original EM | Original F1 |
> |------|----|----|-------------|-------------|
> | bi+ce | 0.1440 | 0.2268 | 0.1335 | 0.2105 |
> | bi+SetCE | **0.1481** | **0.2386** | 0.1513 | 0.2381 |
> | bi+ParaSet | 0.1317 | 0.2131 | 0.0998 | 0.1647 |
> | bi+ParaSet+SetCE | 0.1442 | 0.2266 | 0.0894 | 0.1598 |
>
>
> ---
>
>
> **Discussion**
>
>
> These results show that:
> - Simply increasing CE retrieval depth does not match set-based performance.
> - Combining CE with set-level retrievers consistently yields the best results.
>
>
> This indicates that the two approaches are complementary, helping address different query structures (e.g., sequential vs. parallel).

---

> > ### Author Rebuttal · Reviewer_ntpG · 2026-04-01
> >
> > Thank you. This response reinforces my confidence in my strong evaluation of this interesting work. (I can't justify raising the score or confidence further, but I will maintain them.)

---

> > > ### Author Response · Authors · 2026-04-08
> > >
> > > Thank you for your positive feedback and for taking the time to carefully consider our rebuttal.
> > > We are glad that our clarifications helped address your concerns.
> > > We will further refine the presentation in the final version to make these points clearer.

---

### Decision · Program_Chairs · 2026-04-30

**Decision:**

Reject

**Comment:**

Multi-hop QA requires sets of passages with joint evidence, yet existing methods fail to model set-level compatibility. This paper introduces ParaSet, featuring two contributions: (1) SetCE, a cross-encoder trained with set-level contrastive learning to identify compatible passage sets, and (2) a lightweight Parallel-Set Scorer using self-attention for efficient beam search over combinatorial candidate spaces. At inference, the system retrieves initial passages via bi-encoder, explores combinations with the scorer, and re-ranks with SetCE. Evaluation on HotpotQA, 2WikiMultiHopQA, and MuSiQue confirms its efficiency.

While the reviewers acknowledged that the manuscript addresses a well-motivated and relatively unexplored problem regarding set-level compatibility modeling in multi-hop retrieval, and found the SetCE objective conceptually significant, they expressed substantial concerns regarding the technical soundness and the veracity of the claimed contributions.

A primary point of contention is that the core practical component, the standalone ParaSet, demonstrates inferior performance compared to the standard Bi+CE baseline it intended to surpass. Although the rebuttal characterizes ParaSet as a complementary addition to the Cross-Encoder rather than a substitute, this perspective was not present in the initial submission and indicates a notable alteration in the scope of the contribution. Furthermore, the practical feasibility of selecting an optimal combination is limited by the fact that the query type is generally not known a priori.

During the rebuttal period, the observed performance degradation on the MuSiQue dataset intensifies when utilizing a more advanced retriever with the Qwen3-embedding and different scaling behaviors with larger candidate set, which challenges the claims of the method's practical generalizability. Finally, the comparison with SETR, while showing promise, lacked proper experimental controls and was omitted from the original manuscript.